# No One Size Fits All: QueryBandits for LLM Hallucination Mitigation

## Abstract

Advanced reasoning capabilities in Large Language Models (LLMs) have led to more frequent hallucinations; yet most mitigation work focuses on open-source models for post-hoc detection and parameter editing. The dearth of studies focusing on hallucinations in closed-source models is especially concerning, as they constitute the vast majority of models in institutional deployments. We introduce **QueryBandits**, a model-agnostic contextual bandit framework that adaptively learns online to select the optimal query-rewrite strategy based on a 17-dimensional vector of linguistically motivated features. Evaluating our method on GPT-4o in black-box conditions across 16 QA scenarios, our top QueryBandit (Thompson Sampling) achieves an 87.5% win rate over a NO-REWRITE baseline and outperforms zero-shot static policies (e.g., PARAPHRASE or EXPAND) by 42.6% and 60.3%, respectively. Moreover, all contextual bandits outperform vanilla bandits across all datasets, with higher feature variance coinciding with greater variance in arm selection. This substantiates our finding that there is *no single rewrite policy* optimal for all queries. We also discover that certain static policies incur higher cumulative regret than NO-REWRITE, indicating that an inflexible query-rewriting policy can worsen hallucinations. Thus, learning an online policy over semantic features with QueryBandits can shift model behavior purely through forward-pass mechanisms, enabling its use with closed-source models and bypassing the need for retraining or gradient-based adaptation.

## 1 Introduction

As Large Language Models (LLMs) grow more powerful, the severity of factual errors, otherwise known as *hallucinations*, can increase (OpenAI, 2025; Times, 2025). Hallucinations refer to the generation of inaccurate outputs relative to the LLM's internal *understanding* of the query and reference context (Ji et al., 2023). However, most existing mitigation approaches, especially those relying on logits, token-level probabilities, or internal representation editing, are primarily developed for open-weight models (Touvron et al., 2023)–even though closed-source models constitute the majority of institutional deployments in today's society (OpenAI et al., 2024). Moreover, small surface-form perturbations to an input can induce large output differences (Watson et al., 2025a; Cho & Watson, 2025), underscoring the need for an online, model-agnostic policy-learning process to mitigate hallucinations.

We propose **QueryBandits**, a contextual bandit framework that selects, per query, an appropriate rewrite strategy to proactively steer LLMs away from hallucinations. Interventions are derived from the semantic features, or *fingerprint*, of a query. To formalize the relationship between a query and factuality, we construct a composite reward function, $r_t = \alpha\, s_{\text{llm}} + \beta\, s_{\text{fuzz}} + \gamma\, s_{\text{bleu}}$, where $s_{\text{llm}} \in \{0, 1\}$ is an LLM-as-a-judge binary correctness label (Liu et al., 2023c; Adlakha et al., 2024), $s_{\text{fuzz}} \in [0, 1]$ is a fuzzy string-similarity score (Bachmann, 2024), and $s_{\text{bleu}} \in [0, 1]$ is the BLEU-1 score capturing unigram lexical overlap (Papineni et al., 2002; Callison-Burch et al., 2006). We *operationalize* hallucination as responses with low $r_t$. Through our ablations, we identify the Pareto-optimal balance of weights $(\alpha, \beta, \gamma) = (0.6, 0.3, 0.1)$ on a held-out human labeled set (Fig. 2a). This proxy $r_t$ separates correct from incorrect answers with ROC–AUC 0.973 (95% CI: [0.972, 0.975]) across resampling settings, supporting its use as a learning signal. We assign a higher weight to the LLM-as-a-judge term ($\alpha$) within the Pareto frontier, consistent with studies that highlight the efficacy of LLMs in Natural Language Generation (NLG) evaluation tasks (Wang et al., 2023a; Fu

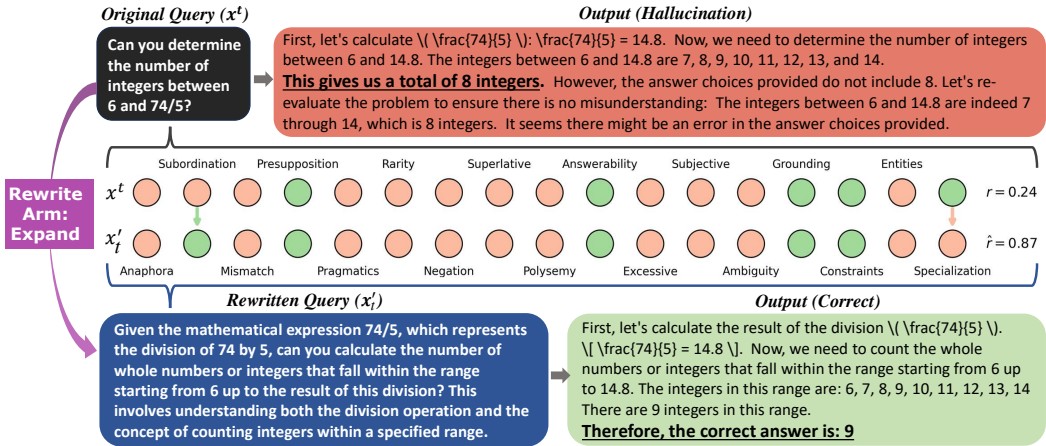

Figure 1: **QueryBandits selects a rewrite that fixes a counting error.** The original query $x_t$ elicits a hallucinatory count (8 integers) due to an ambiguous lower bound (6). Conditioned on the query's 17-dimensional feature vector, QueryBandits selects EXPAND and rewrites the query to $x'_t$ with explicit bounds; the LLM then returns the correct cardinality (9). Noticeably, the feature vector also shifts: *subordination* (more complex clauses) appears while *specialization* (domain-specific knowledge required) disappears-illustrating how rewriting alters the salient semantics of $x_t$.

et al., 2023). We make no stationarity assumption about the reward distribution given the extreme dimensionality of the output space (Riemer et al., 2022), and therefore evaluate whether rewrite strategies confer advantages under both average-reward and worst-case objectives.

Reinforcement Learning (RL) (Sutton & Barto, 2018) methods have been applied in Natural Language Processing (NLP) for tasks such as optimizing document-level retrieval (Nogueira & Cho, 2017), fine-tuning LLMs (Christiano et al., 2017), and post-training (Mudgal et al., 2024). Despite its prevalence, to our knowledge there is limited in-depth research on interactive rewriting for hallucination mitigation. We adopt bandits rather than full RL for three reasons: (i) estimating long-horizon value for hallucination incidence would require repeated queries from a shared subpopulation, whereas interactions are predominantly single-shot; (ii) averaging correctness across heterogeneous contexts obscures informative per-query idiosyncrasies; and (iii) modeling token-level transition dynamics is unwarranted for our objective. That is not to say bandit-style ideas are not without precedent in NLP: Proximal Policy Optimization (Schulman et al., 2017) variants for LLMs such as Group Relative Policy Optimization (GRPO) (Shao et al., 2024) and ReMax (Li et al., 2024c) remove the critic via grouped Monte Carlo or baseline-adjusted returns.

**Action Space and Context.** We define five rewrite strategies as our action space and a 17-dimensional linguistic feature vector capturing query properties known to affect model understanding (Table 10). QueryBandits therefore learns an online policy mapping this validated linguistic feature vector to arm selections, allocating exploration under uncertainty and exploitation when features are predictive. This contrasts with prior approaches that adopt a one-size-fits-all rewrite strategy and do not learn an adaptive selection policy (Ma et al., 2023; Watson et al., 2025a). Our aim is not to propose a new mechanistic theory of hallucination formation, but to cast the rewrite-selection problem as a contextual bandit with bounded rewards. Under this view, the bandit's optimal policy minimizes expected hallucination probability as proxied by our reward. Existence of such a policy follows from standard bandit theory under bounded rewards, and our empirical analyses show that Thompson Sampling and LinUCB converge toward high-reward rewrite policies in our setting (Auer et al., 2002a; Lattimore & Szepesvári, 2020).

**Contribution 1: Reward Modeling for Factuality.** We introduce an empirically validated and calibrated reward function $r_t$, composed of an LLM-judge, fuzzy-match, and BLEU-1 metrics, with $\alpha, \beta, \gamma = (0.6, 0.3, 0.1)$ chosen inside the 1% Pareto-optimal frontier on a held-out human-labeled set (Fig. 2a). Our evaluation rests on the simplex formed by $\alpha, \beta, \gamma \geq 0$, $\alpha + \beta + \gamma = 1$. The reward reliably separates right from wrong answers: its average ROC–AUC is 0.973 across resampling settings, and even the conservative 95% lower bound exceeds 0.97 after 150 samples, indicating a

Table 1: **Accuracy by dataset (rows) and algorithm family (columns).** Higher is better; **bold** marks the row maximum. "Wins (ties split)" counts 0.5 for ties. "Macro-avg" is the unweighted mean across datasets. Contextual methods dominate: Contextual Thompson Sampling (TS, right-most column) achieves the best macro-average (0.766) and most wins (8/16); the remaining wins come from the linear contextual family (LinUCB 4.5, LinUCB+KL 3.5). Static prompts and non-contextual bandits do not win on any dataset. NoRw = No-Rewrite.

| Dataset | Base | Static Prompts | | | | | Non-Contextual | | | | Contextual Linear | | | | |
|---|---|---|---|---|---|---|---|---|---|---|---|---|---|---|---|
| | NoRw | Para. | Simpl. | Disamb. | Clarify | Expand | EXP3 | FTPL | $\epsilon$-FTRL | TS (NC) | LinUCB | LinUCB+KL | LinEXP3 | LinFTPL | TS (C) |
| **ARC-Challenge** | 0.816 | 0.813 | 0.814 | 0.786 | 0.800 | 0.731 | 0.878 | 0.792 | 0.873 | 0.887 | **0.888** | **0.888** | 0.878 | 0.826 | 0.884 |
| **ARC-Easy** | 0.808 | 0.807 | 0.810 | 0.796 | 0.793 | 0.748 | 0.890 | 0.743 | 0.859 | 0.877 | 0.892 | 0.888 | 0.869 | 0.818 | **0.895** |
| **BoolQA** | 0.547 | 0.564 | 0.574 | 0.574 | 0.568 | 0.554 | 0.658 | 0.589 | 0.649 | 0.571 | 0.649 | 0.668 | 0.637 | 0.605 | **0.673** |
| **HotpotQA** | 0.658 | 0.653 | 0.657 | 0.664 | 0.650 | 0.654 | 0.755 | 0.660 | 0.747 | 0.667 | **0.764** | 0.757 | 0.726 | 0.670 | 0.756 |
| **MathQA** | 0.700 | 0.692 | 0.678 | 0.685 | 0.689 | 0.691 | 0.779 | 0.688 | 0.758 | 0.756 | **0.787** | 0.784 | 0.732 | 0.696 | 0.785 |
| **MMLU** | 0.744 | 0.748 | 0.724 | 0.736 | 0.728 | 0.709 | 0.832 | 0.747 | 0.803 | 0.773 | **0.837** | 0.832 | 0.780 | 0.721 | 0.835 |
| **OpenBookQA** | 0.735 | 0.736 | 0.738 | 0.677 | 0.667 | 0.553 | 0.769 | 0.725 | 0.776 | 0.780 | 0.790 | 0.791 | 0.718 | 0.694 | **0.793** |
| **PIQA** | 0.717 | 0.715 | 0.729 | 0.639 | 0.666 | 0.561 | 0.772 | 0.638 | 0.755 | 0.733 | 0.785 | **0.791** | 0.766 | 0.746 | 0.790 |
| **SciQ (Abstract)** | 0.712 | 0.725 | 0.701 | 0.706 | 0.704 | 0.680 | 0.804 | 0.704 | 0.773 | 0.780 | 0.800 | 0.802 | 0.725 | 0.693 | **0.806** |
| **SciQ (MC)** | 0.775 | 0.777 | 0.771 | 0.766 | 0.749 | 0.704 | 0.847 | 0.764 | 0.823 | 0.828 | 0.851 | 0.857 | 0.796 | 0.787 | **0.867** |
| **SQuAD (Abstract)** | 0.531 | 0.559 | 0.540 | 0.540 | 0.531 | 0.507 | 0.626 | 0.553 | 0.614 | 0.523 | 0.632 | 0.628 | 0.606 | 0.568 | **0.636** |
| **SQuAD (Extract)** | 0.670 | 0.679 | 0.681 | 0.643 | 0.640 | 0.565 | 0.742 | 0.682 | 0.738 | 0.682 | 0.743 | 0.752 | 0.748 | 0.697 | **0.759** |
| **TriviaQA** | 0.682 | 0.668 | 0.662 | 0.651 | 0.646 | 0.653 | 0.742 | 0.670 | 0.734 | 0.729 | 0.754 | **0.759** | 0.693 | 0.671 | 0.757 |
| **TruthfulQA** | 0.496 | 0.488 | 0.509 | 0.481 | 0.470 | 0.441 | 0.567 | 0.509 | 0.577 | 0.516 | 0.583 | **0.595** | 0.555 | 0.512 | 0.586 |
| **TruthfulQA (MC)** | 0.807 | 0.791 | 0.834 | 0.753 | 0.741 | 0.679 | 0.854 | 0.705 | 0.802 | 0.887 | **0.888** | 0.863 | 0.846 | 0.786 | 0.852 |
| **WikiQA** | 0.498 | 0.494 | 0.498 | 0.472 | 0.485 | 0.470 | 0.581 | 0.519 | 0.562 | 0.566 | 0.570 | 0.576 | 0.557 | 0.514 | **0.590** |
| **Macro-avg** | 0.681 | 0.682 | 0.682 | 0.661 | 0.658 | 0.619 | 0.756 | 0.668 | 0.740 | 0.722 | 0.763 | 0.764 | 0.727 | 0.688 | **0.766** |
| **Wins (ties split)** | – | – | – | – | – | – | – | – | – | – | 4.5 | 3.5 | – | – | **8.0** |

stable and highly discriminative proxy for correctness. Guided by this reward signal, our contextual QueryBandits learn to tailor rewrite choices to each query's linguistic/contextual fingerprint.

**Contribution 2: Contextual Adaptation Wins.** Across 13 QA benchmarks (16 scenarios), our best contextual bandit, Thompson Sampling (TS), drives an 87.5% win rate over the NO-REWRITE baseline and outperforms zero-shot static policies (PARAPHRASE, EXPAND) by 42.6% and 60.3%, respectively. Furthermore, certain static strategies accrue higher cumulative regret than NO-REWRITE, indicating that *fixed rewrites can worsen hallucination*. In Fig. 3, contextual QueryBandits quickly hone in on the optimal rewrites, accruing substantially lower cumulative regret than static policies, vanilla (non-contextual) bandits, or no-rewriting. These gains confirm that a feature-aware, online adaptation mechanism consistently outpaces one-shot heuristics in mitigating hallucinations.

**Contribution 3: Interpretable Decision Weights.** Per-arm regression analyses (Fig. 5) provide empirical evidence that *no single rewrite strategy* maximizes the reward across all types of queries. In fact, each arm's effectiveness hinges on the semantic features of a query. For example, if a query displays the feature *(Domain) Specialization*, meaning that the query can only be understood with domain-specific knowledge, the rewrite arm EXPAND is very effective in contrast to SIMPLIFY (Figure 1). Ablating the 17-feature context reduces TS's win rate to 81.7% and the exploration-adjusted reward to 754.66. Macro-averaged accuracy across the 16 scenarios corroborates this decline: non-contextual TS drops to 72.2% from 76.6%. This performance gap confirms that linguistic features carry associative signals about the optimal rewrite strategy. To our knowledge, this is the first work to use a holistic 17-feature linguistic vector as per-query context for a bandit's best-arm selection– moving beyond piecemeal correlations to a single-pass, end-to-end decision policy. Finally, we observe that across datasets, higher feature variance coincides with greater variance in arm selection (Figure 4), yielding genuinely diverse arm choices (Figure 2b).

**Contribution 4: Scope & Utility.** QueryBandits operates entirely at the input layer as a model-agnostic, plug-and-play online learning policy suitable for closed-source LLMs, addressing the critical arena of hallucination mitigation efforts where model weights are inaccessible. By contrast, existing mitigation methods for open-source models such as DoLa (Chuang et al., 2024) and TruthX (Zhang et al., 2024a) modify internal representations or decoding, neither of which are directly available for closed models. On TRUTHFULQA (Lin et al., 2022), their gains on smaller open models (LLAMA-2-7B-CHAT) remain far below strong closed backbones (TruthX: **54.2%**, DoLa: **32.2%**, vs. GPT-4o: **81.4%**). QueryBandits further lifts GPT-4o from 81.4% to **88.8%** MC1 (+7.4 pp) by adapting rewrites to per-query features, with minimal compute and token overhead. Because DoLa/TruthX gains are realized on weaker open models, they do not transfer additively at higher baselines due to diminishing headroom.

**Interesting Findings.** (i) On many standard benchmarks, linear contextual bandits often converge to the NO REWRITE arm (Figure 8), exposing memorization effects. Diversity emerges only when queries are semantically invariant but lexically perturbed; a meaningful insight for the research

community that surface-form novelty is essential in training query-rewriting algorithms. (ii) Non-contextual bandits often converge to a single rewrite strategy per dataset, whereas contextual bandits tend to diversify choices conditioned on the presence and/or absence of linguistic features.

**Key Empirical Takeaway.** Taken together, the dominance of contextual learners, the consistent edge of non-contextual bandits over static prompts, and the near-parity of static prompts with the NO-REWRITE baseline indicate that (a) per-query linguistic features reliably predict rewrite utility, (b) online adaptation matters even without features, and (c) there is no universally beneficial fixed policy on strong LLMs (Tables 1, 4).

## 2 RELATED WORKS

**Societal Stakes and Gap of Closed-Source Models.** LLM hallucinations erode trustworthiness from a societal perspective (Dechert LLP, 2024). Recent conceptual analyses frame them as a new epistemic failure mode requiring dedicated mitigation agendas (Yao et al., 2024). Complementing these views, Kalai et al. (2025) argue that language models hallucinate because prevailing training and evaluation procedures reward guessing over acknowledging uncertainty. Reports on newer advanced-reasoning models (e.g., O3, O4-MINI) indicate increased hallucination rates (OpenAI, 2025), and journalistic case studies document real-world legal exposure from fabricated outputs (Times, 2025). As more LLM-agent systems proliferate (Watson et al., 2025b; 2023), the downstream cost of errors compounds. Yet, there remains a dearth of studies on hallucination mitigation efforts for *closed-source* models–our work targets this underexplored gap (Huang et al., 2025b; Tonmoy et al., 2024; Sahoo et al., 2025).

**From Post-hoc Detection to Preemptive Query Shaping.** Mitigation is indispensable for faithful LLM interaction (Ji et al., 2023), and research has expanded from post-hoc detection and iterative correction (Madaan et al., 2023) to preemptive grounding and query restructuring. Watson et al. (2025a) estimate hallucination risk *before* generation via query perturbations. Ma et al. (2023) propose *Rewrite-Retrieve-Read* for RAG pipelines, and manual, rule-based rewriting is widely used (Liu & Mozafari, 2024; Mao et al., 2024; Chen et al., 2024a). A common limitation is reliance on raw prompting or static heuristics rather than *guided* rewrites conditioned on the original query's contextual signals.

**Linguistic Features as Actionable Context.** Blevins et al. (2023) show that pretrained language models can recover linguistic attributes in a few-shot setting. Building on this, we employ an LLM to identify 17 key linguistic features per query (Table 10). Feature selection drew from both existing LLM literature and traditional linguistics, prioritizing properties known to affect comprehension for humans and models alike. These features serve as the context for our bandit policy, enabling *feature-conditioned* query-rewriting rather than one-size-fits-all rules.

## 3 METHODOLOGY AND EVALUATION METRICS

**Bandit Formulation.** In the contextual multi-armed bandit framework (Lattimore & Szepesvári, 2020), a learner observes at round $t$ a context vector $x_t \in \mathcal{X} \subset \mathbb{R}^d$ and selects an arm $a_t \in \mathcal{A}$. Upon that basis, Nature reveals a scalar reward $r_t = r(x_t, a_t) \in [0, 1]$, where $r : \mathcal{X} \times \mathcal{A} \to [0, 1]$. The goal of a bandit algorithm is to select arms that maximize the expected (cumulative) reward (Alg. 1; Appx. D). In the stochastic bandit setting, the objective is to choose a *policy* $\pi : \mathcal{X} \to \rho(\mathcal{A})$ that maximizes the expected reward, i.e.,

$$\max_{\pi \in \Pi} \mathbb{E}\left[ r(x, \tilde{a}) \right], \quad \tilde{a} \sim \pi(x),$$

where $\rho(\mathcal{A})$ is the probability simplex over $K = |\mathcal{A}|$ arms, and $\Pi$ is the policy class.

**Action Space.** Let $\mathcal{A} = \{a_0, \dots, a_{K-1}\}$ denote the rewrite strategies (arms), where each $a_i \in \mathcal{A}$ represents a distinct style of query reformulation implemented via prompt instructions to an LLM:

▶ $a_0$ PARAPHRASE: Rewrite the query to introduce lexical diversity while preserving semantic meaning, testing whether alternative phrasings reduce hallucinations. Prior work has explored how paraphrasing can improve factual consistency in LLMs (Deng et al., 2024; Witteveen & Andrews, 2019).

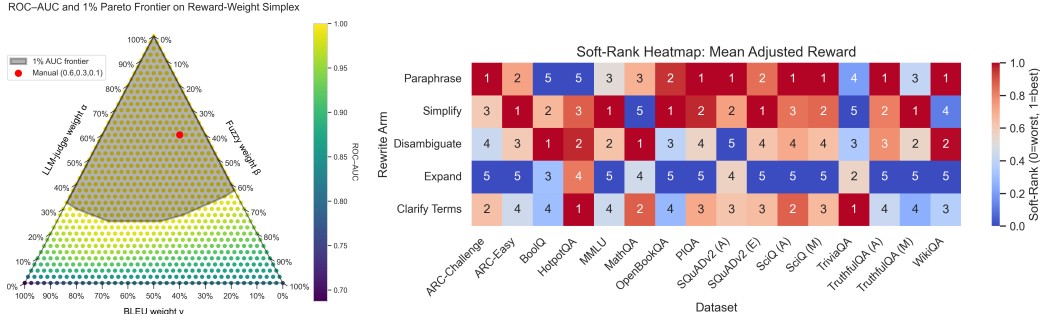

(a) ROC–AUC Pareto frontier on the reward-weight simplex.

(b) Mean-reward ranks (1 = best) per rewrite arm / dataset under our contextual bandit; color intensity indicates closeness to the top rank.

Figure 2: (a) Our chosen $(\alpha, \beta, \gamma)$ lies deep in the 1% optimal frontier. (b) Breakdown of per-dataset arm performance: different datasets consistently favor different rewrite strategies

▶ $a_1$ SIMPLIFY: Rewrite the query to eliminate nested clauses and complex syntax. This targets hallucinations caused by long-range dependencies or overloaded details, borrowing ideas from educational psychology where simpler, granular prompts enable a child to learn a new skill (Libby et al., 2008). Recently, Van et al. (2021); Zhou et al. (2023) report that simplified prompts reduce off-topic drift and ease reasoning.

▶ $a_2$ DISAMBIGUATE: Rewrite the query by resolving vague references (ambiguous pronouns, temporal expressions). Studies showcase LLMs' inability to resolve ambiguous queries, leading to subpar performance (Deng et al., 2023; Shahbazi et al., 2019). The information required to disambiguate is obtained by rephrasing and making implicit references explicit *using only the original query context*, without relying on external knowledge.

▶ $a_3$ EXPAND: Rewrite the query to add salient entities and attributes to enrich context (Yu et al., 2023). Since transformers optimize next-token likelihood over attention-mediated context windows (Vaswani et al., 2023), appending fine-grained query constraints effectively conditions the model on a richer semantic prefix.

▶ $a_4$ CLARIFY TERMS: Rewrite the query to define jargon and terms of art to reduce domain-specific ambiguity (Clark & Gerrig, 1983; Rippeth et al., 2023). This is especially useful for *long-tail knowledge*, where LLMs underperform on less-popular entities and benefit from added context or lightweight retrieval (Mallen et al., 2023).

In our experiments, we instantiate all rewrite arms using `gpt-4o-2024-11-20`; stronger (or weaker) rewriters can be substituted without changing the bandit formulation.

**Contextual Attributes.** For each query we extract a 17-dimensional binary feature vector $\mathbf{f} \in \{0, 1\}^{17}$ capturing linguistically motivated properties known to affect human and LLM comprehension (Table 10). These features serve as the context for our policy, giving contextual bandits the opportunity to learn *when* to apply which rewrite.

**Reward Model.** Each rewritten query receives a bounded composite reward $r_t \in [0, 1]$ as a convex combination of three complementary correctness signals:

$$r_t = \alpha \, s_{\text{llm}} + \beta \, s_{\text{fuzz}} + \gamma \, s_{\text{bleu}}, \qquad \alpha + \beta + \gamma = 1, \quad \alpha, \beta, \gamma \geq 0 \tag{1}$$

▶ $s_{\text{llm}} \in \{0, 1\}$: a binary correctness judgment by a GPT-4o-based assessor, calibrated on factuality between generated and reference answers (Liu et al., 2023c; Adlakha et al., 2024).

▶ $s_{\text{fuzz}} \in [0, 1]$: RapidFuzz token-set similarity capturing soft string overlap (Bachmann, 2024).

▶ $s_{\text{bleu}} \in [0, 1]$: BLEU-1 (unigram precision) under a unit-cap ensuring lexical fidelity (Papineni et al., 2002; Callison-Burch et al., 2006).

This triad mitigates individual failure modes inherent in any single metric (e.g. BLEU's paraphrase blindness or edit-distance oversensitivity) while remaining stable for learning. Following Wang et al. (2023a), we leverage the strength of LLMs-as-judges; and as demonstrated by Test-Time RL (Zuo et al., 2025), even noisy, self-supervised signals (e.g. pseudo-labels from majority-voted LLM outputs) can effectively guide policy updates. We validate that our convex proxy $r_t$ aligns with human labels via a 1,000 sample held-out set and report ROC-AUC in Figures 2a and 6.

**Validity of the Reward & Simplex Analysis.** Across sample sizes (5–1000 samples), the reward attains macro-average ROC–AUC **0.9729**; by 150 samples the 95% CI lower bound exceeds 0.97, indicating a stable and highly discriminative correctness proxy (Fig. 6b; Tab. 6a). We sweep $(\alpha', \beta', \gamma')$ over a simplex grid ($\alpha' + \beta' + \gamma' = 1$) and computed ROC–AUC on the human-labeled validation set (Fig. 2a). Our best weights $(\alpha, \beta, \gamma) = (0.6, 0.3, 0.1)$ **lie** well within the top 1% Pareto frontier (dark region) and is robust to $\pm 0.2$ perturbations on $\alpha$. The Pareto frontier reveals the following:

- ▸ **LLM-Judge Robustness ($\alpha$):** The ROC–AUC surface is nearly invariant when $\alpha$ varies by $\pm 0.2$: AUC shifts by $< 0.5\%$, indicating tolerance to large $\alpha$ swings.
- ▸ **Fuzzy-Match Sensitivity ($\beta$):** Small increases in $\beta$ rapidly exit the Pareto region, showing that the fuzzy-match term must be tuned carefully to avoid degrading overall accuracy.
- ▸ **BLEU-Only Pitfall ($\gamma$):** As $\gamma$ increases, ROC–AUC steadily declines, bottoming at $\gamma = 1$ (pure-BLEU), where the model over-emphasizes surface overlap at the expense of true correctness.
- ▸ **Pareto-Optimal Region:** The weights $(0.6, 0.3, 0.1)$ sit deep in the high-AUC plateau, confirming it is a Pareto-optimal trade-off among semantic, fuzzy, and lexical signals.
- ▸ **Reward Non-degeneracy ($\beta, \gamma$):** Using only the LLM-Judge term ($\alpha = 1$) yields a nearly binary reward distribution that collapse onto two modes, which in turn hurts exploration-exploitation. Adding the fuzzy and BLEU terms yields richer, more graded rewards that are sensitive to *near misses* (Fig. 14)

Together, these experiments substantiate our reward design: the LLM-judge provides a forgiving anchor, fuzzy-match demands precise calibration, and BLEU contributes complementary lexical oversight. We further evaluated reward robustness with out-of-family judges (`gpt-5*`, `gpt-4.1-2025-04-14`, and `gpt-4o*`). Across 1,000 validation queries, inter-model agreement on correctness labels is high (mean agreement $\approx 0.9$, mean $\kappa \approx 0.79$, MCC $\approx 0.80$), indicating that our reward is stable across judge architectures (Table 6).

**Choice of Algorithms.** For **linear contextual bandits**, we fit a per-arm linear model $x_t^\top \theta_k$ and use either a UCB method (LinUCB (Lai & Robbins, 1985) / LinUCB+KL (Garivier & Cappé, 2013)), an FTRL regularized weight (McMahan, 2015), or Thompson sampling with posterior draws (Thompson, 1933). For **adversarial bandits**, we consider two parameter-free methods: EXP3 (Auer et al., 2002b) and FTPL (Kalai & Vempala, 2005; Suggala & Netrapalli, 2020). Update rules and regret bounds are in App. D (Alg. 1). We discuss our decision to use bandits rather than full RL in App. B.

**Evaluation Metrics.** We report three complementary metrics for a balanced view of (1) how well a policy explores vs. exploits, (2) how quickly it converges to good answers, and (3) how often it beats the NO-REWRITE baseline in accuracy.

**Metric 1: Exploration-Adjusted Reward.** Let $r_t \in [0, 1]$ be the reward at pull $t$ up to trajectory length $T$. Define the empirical arm-frequency vector $p_{t,k} = \frac{1}{t} \sum_{\tau=1}^{t} \mathbf{1}[a_\tau = k]$ and the normalized Shannon entropy $H_t = \left(-\sum_{k=1}^{K} p_{t,k} \log p_{t,k}\right) / \log K \in [0, 1]$. We define the *exploration-adjusted reward* as:

$$R_{\text{adj}} = \sum_{t=1}^{T} \left(r_t + \lambda H_t\right),$$

with $\lambda = 0.1$ (chosen on validation), rewarding policies that achieve high per-pull rewards while maintaining sufficient exploration.

**Metric 2: Mean Cumulative Regret.** At each pull the instantaneous regret is the gap between the oracle reward (best achievable rewrite) and the observed reward. Let $r_t^* = \max_{a \in \mathcal{A}} r(x_t, a)$ be the per-round oracle (max) reward. Over $R$ runs, the mean cumulative regret is:

$$\overline{\text{Regret}} = \frac{1}{R} \sum_{i=1}^{R} \sum_{t=1}^{T} \left(r_t^* - r_t^{(i)}\right)$$

**Metric 3: Win Rate vs. Baseline.** For $N$ test queries, we compute the fraction of trials where a policy's reward $r_t^{\text{policy}}$ strictly exceeds the no-rewrite baseline $r_t^{\text{base}}$:

$$\text{WinRate} = \frac{1}{N} \sum_{t=1}^{N} \mathbf{1}\left[r_t^{\text{policy}} \succ r_t^{\text{base}}\right] \times 100\%.$$

## 4 EXPERIMENTS

**Pipeline.** For each decision round $t$:

$$x_t \xrightarrow[\mathbf{f}_t \in \{0,1\}^d]{\text{Extr. feat.}} \mathbf{f}_t \xrightarrow[\text{(rewrite strat.)}]{\text{Select } a_t} x'_t = g_{a_t}(x_t) \xrightarrow{\text{LLM}} y_t \xrightarrow[r_t \in [0,1]]{\text{Eval.}} r_t \qquad \overset{\text{Update Bandit}}{\hookleftarrow} P$$

1. **Feature Extraction.** For query $x_t$, compute $d$-dimensional linguistic feature vector $\mathbf{f}_t \in \{0,1\}^d$.
2. **Arm Selection.** The bandit receives $\mathbf{f}_t$ and selects a rewrite arm $a_t \in \mathcal{A}$.
3. **Query Rewriting.** Apply the selected arm to obtain the candidate query $x'_t = g_{a_t}(x_t)$.
4. **LLM Inference.** Issue $x'_t$ to `gpt-4o-2024-08-06`, producing response $y_t$.
5. **Reward Evaluation.** Compute scalar reward $r_t \in [0, 1]$ via the reward formulation.
6. **Bandit Update.** Update the internal state of the bandit based on $(a_t, r_t)$.

**Dataset and Query Construction.** We evaluate on $D = 13$ diverse QA benchmarks and $S = 16$ scenarios (see Table 3). For each scenario, we sample $|\mathcal{Q}|$ queries satisfying: (1) *Original Answerability:* the query in the dataset ($q_i$) is answered correctly by `gpt-4o-2024-08-06`; and (2) *Perturbation Validity:* among five lexically perturbed but semantically invariant versions of each dataset query, assessed by an LLM-as-judge and n-gram based metrics (Lin, 2004; Papineni et al., 2002; Wang et al., 2023a), between one and three perturbations yield incorrect answers. Then, we randomly choose $x_t$ from $|\mathcal{Q}|$ to train QueryBandits.

The importance of this query construction process deserves emphasis. Through our investigations, we discovered that the ubiquity of benchmarks in Table 3 within pre-training and fine-tuning regimes has engendered a potentially pernicious form of prompt memorization. In preliminary runs using canonical, unperturbed queries, contextual policies often converge almost exclusively to NO-REWRITE, and rewriting rarely improved accuracy. By contrast, in our perturbed setup (lexically diverse but semantically matched queries), contextual bandits diversify arm usage and achieve substantial gains (Figure 8). This behavior is consistent with prompt memorization on common benchmarks rather than an intrinsic degradation effect of rewriting.

**Experimental Configuration.** We compare three non-contextual and six linear contextual bandits against zero-shot prompting and a NO-REWRITE baseline. All reported metrics are averaged over all dataset runs per algorithm. We compare $M$ bandit algorithms and prompting strategies over $K = 5$ rewrite arms. Each algorithm runs for $T = |\mathcal{Q}_S|$ rounds on each of the $S$ scenarios (Table 3). Thus, Total Pulls $= M \times S \times |\mathcal{Q}_S| \approx 252,000$, with $M = 15$, $S = 16$, and $|\mathcal{Q}_D| \approx 1050$. We bootstrap samples with replacement for TRUTHFULQA to obtain approximately 1,050 queries. Hyperparameters (learning rates, exploration coefficients, regularization constants) are tuned via grid search on a held-out validation set.

**Feature Extraction.** We use `gpt-4o-2024-11-20` with temperature $\tau = 0.0$ and structured outputs to tag the 17 binary linguistic features per query (Table 11). On 1,000 queries $\times$ 5 repeated tagging runs, bitwise agreement across full 17-dimensional vectors is $\sim 99.3\%$, and per-feature stability is 97.4%-99.7%, indicating that the contextual representation is nearly deterministic under our setup. Because the bandit only observes the binary feature vector (and not the text), this residual variance has minimal impact on downstream learning.

## 5 RESULTS

**Hypothesis 1: Can QueryBandits reduce hallucination?** Table 2 and Figure 3 compare Query-Bandits against the NO REWRITE baseline and five static prompting strategies across 13 QA benchmarks (16 scenarios, 1,050 queries/dataset). In aggregate, contextual Thompson Sampling (TS) attains an **87.5% query-level win rate** and 819.04 exploration-adjusted reward, compared to the NO REWRITE baseline (729.20; $\Delta = -89.84$). At the scenario level (Table 1), the macro-average accuracy improves from **0.681** (Baseline) to **0.766** (Contextual TS; $+8.5$ pp). Contextual TS also wins **8/16** scenarios outright (Table 2). Together, these results indicate that *contextual* query rewriting materially reduces hallucination relative to no rewriting.

**Hypothesis 2: Can QueryBandits outperform static rewriting?** Static rewriting never tops a dataset on accuracy (Table 1). Our best performing bandit, Contextual TS, consistently exceeds the performance of static variants; for example, relative to PARAPHRASE and EXPAND, Contextual TS

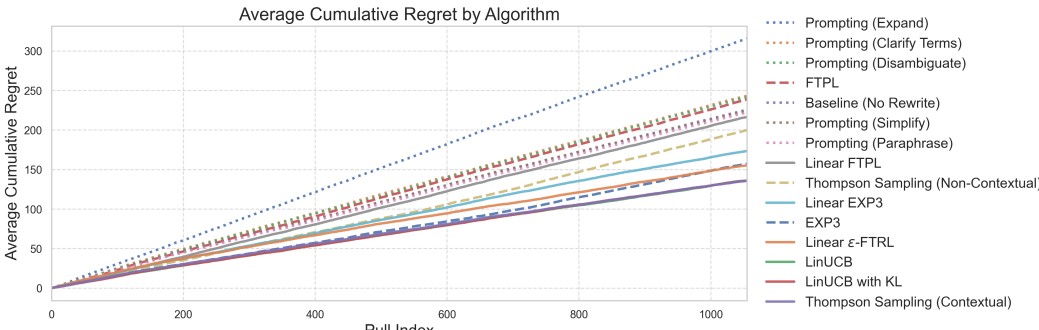

Figure 3: **Cumulative Reward (averaged across all runs).** Sorted by final performance, highlighting gains achieved by contextual bandits over non-contextual learners and static rewrites.

Table 2: **Left: Rewrite-policy Performance:** final cumulative exploration-adjusted reward, mean cumulative regret, and win rate vs. no-rewrite. **Right: Who Wins Where:** best accuracy per dataset and gain over NO-REWRITE baseline (pp). TS = Thompson Sampling; (C) = Contextual.

| Algorithm | Ctx? | $r_{adj}$ ↑ | Cum. Regret↓ | Win% ↑ | | Dataset | Winner Algo. | Acc. (%) ↑ | Δ (pp) ↑ |
|---|---|---|---|---|---|---|---|---|---|
| *Bandit Algorithms* | | | | | | *Winners: TS (Contextual)* | | | |
| **TS (C)** | ✓ | **819.04** | **135.84** | **87.5** | | ARC-Easy | TS (C) | 89.5 | +8.7 |
| LinUCB+KL | ✓ | 818.79 | 136.00 | 87.0 | | BoolQA | TS (C) | 67.3 | +12.6 |
| LinUCB | ✓ | 818.60 | 136.12 | 86.9 | | OpenBookQA | TS (C) | 79.3 | +5.8 |
| Linear $\epsilon$-FTRL | ✓ | 799.57 | 155.30 | 85.0 | | SciQ (Abstract) | TS (C) | 80.6 | +9.4 |
| EXP3 (NC) | ✗ | 797.47 | 157.31 | 86.5 | | SciQ (MC) | TS (C) | 86.7 | +9.2 |
| Linear EXP3 | ✓ | 781.05 | 173.60 | 83.8 | | SQuAD (Abstract) | TS (C) | 63.6 | +10.5 |
| TS (NC) | ✗ | 754.66 | 200.18 | 81.7 | | SQuAD (Extract) | TS (C) | 75.9 | +8.9 |
| Linear FTPL | ✓ | 738.07 | 216.54 | 76.3 | | WikiQA | TS (C) | 59.0 | +9.2 |
| FTPL (NC) | ✗ | 716.05 | 238.85 | 62.8 | | *Winners: LinUCB family* | | | |
| *Static Prompts* | | | | | | ARC-Challenge | LinUCB (+KL) | 88.8 | +7.2 |
| Paraphrase | – | 732.39 | 222.56 | 44.9 | | HotpotQA | LinUCB | 76.4 | +10.6 |
| Simplify | – | 730.13 | 224.42 | 50.1 | | MathQA | LinUCB | 78.7 | +8.7 |
| Disambiguate | – | 713.65 | 241.25 | 42.4 | | MMLU | LinUCB | 83.7 | +9.3 |
| Clarify Terms | – | 711.65 | 243.35 | 38.2 | | PIQA | LinUCB+KL | 79.1 | +7.4 |
| Expand | – | 639.25 | 315.71 | 27.2 | | TriviaQA | LinUCB+KL | 75.9 | +7.7 |
| | | | | | | TruthfulQA | LinUCB+KL | 59.5 | +9.9 |
| No-Rewrite (B) | – | 729.20 | 225.85 | – | | TruthfulQA (MC) | LinUCB | 88.8 | +8.1 |

achieves much higher aggregate reward (819.04 vs. 732.39 and 639.25) and substantially higher accuracy, with typical gains of **+6–12 pp** over the baseline across scenarios (Table 2; e.g., +12.6 on BoolQA, +10.6 on HotpotQA). In 13/15 runs, non-contextual bandits effectively collapse to a single rewrite per dataset, behaving similarly to static policies. In contrast, contextual policies maintain more diverse selections conditioned on feature patterns (Fig. 7). These gains confirm that adapting the rewrite to each query's linguistic fingerprint outperforms any one-size-fits-all prompt. By framing rewrite selection as an online decision problem and leveraging per-query context, QueryBandits allocate exploration where uncertainty is high and exploitation where features reliably predict hallucination risk–yielding up to double the hallucination reduction of any static strategy, with no additional model fine-tuning.

**Hypothesis 3: Do linear contextual bandits outperform algorithms oblivious to context?** Crucially, ablating the 17-dimensional feature vector drops Thompson Sampling's performance from 87.5% to **81.7%** query-level win rate and from 819.04 to **754.66** reward ($-5.8$ pp, $-64.38$ reward). On accuracy, *contextual* methods dominate: Thompson Sampling wins 8/16 scenarios, while the contextual linear family (LinUCB/LinUCB+KL) takes the rest (tie-split: *LinUCB 4.5*, *LinUCB+KL 3.5*); see Table 1. Non-contextual bandits never top accuracy on any dataset. On regret (Table 4), wins spread to simpler methods–NO REWRITE (BASELINE) (3 scenarios), PARAPHRASE (3.5), SIMPLIFY (2), and Non-Contextual TS (3.5)–while contextual methods rarely minimize *instantaneous* regret (only LinFTPL wins once). This pattern aligns with exploration–exploitation: contextual learners accept small exploration costs (slightly higher regret early) to deliver higher final accuracy. While EXP3 is a strong non-contextual baseline, contextual TS stochastically dominates both EXP3 and static policies in per-query reward (Figs. 15–18). This confirms that the gains we observe stem from genuine contextual adaptation rather than noise. Furthermore, these performance gaps confirm that linguistic features carry associative signals about hallucination risk.

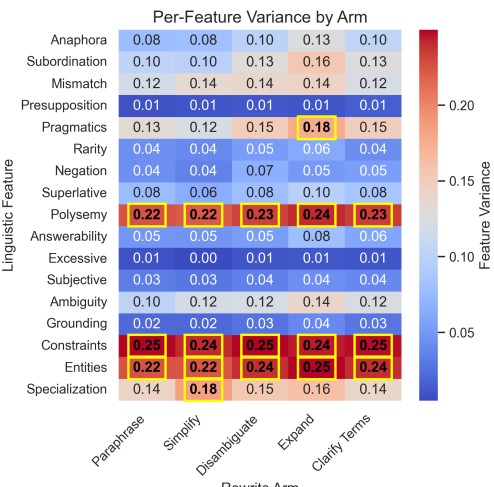
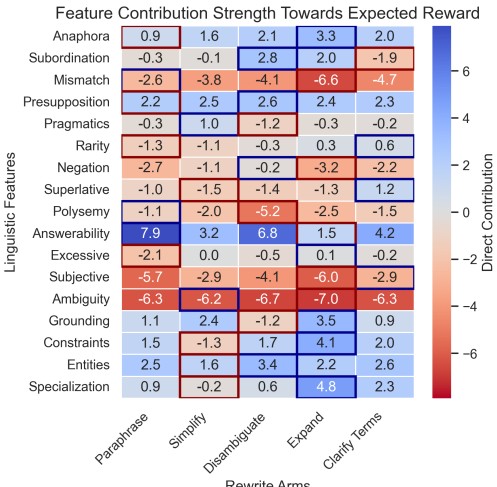

Figure 4: **Contextual Per-Feature Variance by Arm.** For each arm, we compute the variance of each binary linguistic feature over all queries on which that arm was chosen. High variance means the bandit frequently switches the arm on that feature's presence.

Figure 5: **Contextual Feature Contribution Strength.** These are the averaged $\theta$ weights (direct contributions) per feature to the expected reward under each arm. Positive weights indicate features that boost that arm's reward; negative weights indicate features that *penalize* it.

**Hypothesis 4: Is there an association between query features and reward?** Arms exhibit distinct sensitivities to the 17 linguistic features (Figures 4–5). The same feature can flip importance across arms; e.g., *(Domain) Specialization* is highly predictive for EXPAND but weak for SIMPLIFY. A plausible mechanism is that domain-specific questions need added qualifiers/entities (EXPAND) to ground retrieval and reasoning, whereas aggressive pruning (SIMPLIFY) risks *excising* critical semantics. These arm–feature associations are correlational rather than causal, but they are consistent with the observed accuracy/regret trade-offs.

**Hypothesis 5: Is there a *single* rewrite strategy that maximizes reward for all types of queries?** No. The learned per-arm weights (Figure 5) show distinct *feature fingerprints*. For instance, SIM-PLIFY excels with pragmatic cues (safe pruning) but struggles on superlatives (removing comparative meaning). Appendix Table 8 details these inversions. The diversity of winning arms across scenarios (Table 2) and the split of contextual winners (Contextual TS vs. LinUCB family) further support that *no single rewrite strategy is universally optimal*.

**Hypothesis 6: Does QueryBandits improve closed-source model performance?** As shown in Table 5, methods such as DoLa and TruthX improve *open-source* backbones (e.g., Llama-2-7B-Chat), but their best reported MC1 (TruthX: 54.2%; DoLa: 32.2%) is far below strong *closed-source* backbones (GPT-4o: 80.7%) (Zhang et al., 2024a; Chuang et al., 2024). By contrast, QueryBandits operates entirely at the input layer and lifts GPT-4o to 88.8% (+8.1 pp). Since DoLa/TruthX modify internal representations or decoding, they are not directly applicable to closed models, and gains on weaker models need not transfer additively at higher baselines.

## 6 CONCLUSION

We introduce QUERYBANDITS, a plug-and-play online learning policy that selects among $K$ rewrite strategies to minimize a query's *hallucinatory* trajectory using lightweight linguistic features as context. Across 13 QA benchmarks (16 scenarios), *contextual* learners dominate: Contextual TS and the LinUCB family win nearly all benchmarks, yielding a macro-average accuracy of **0.766** vs. **0.681** for NO-REWRITE, with typical gains of ∼6–12 pp (Table 1). Non-contextual bandits generally beat static prompts, while static prompts are on par with the baseline, indicating that (i) per-query features predict rewrite utility, (ii) online adaptation matters even without features, and (iii) no single fixed rewrite is universally beneficial on strong LLMs.

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

# A APPENDIX

## A.1 LIMITATIONS

Current limitations in our work are as follows: our current contextual bandit framework treats each of the 17 features as independent, but does not capture higher-order interactions. This can provide an exciting avenue of future research in terms of measuring whether the combination of features jointly exacerbates hallucination. Likewise, we would like to highlight that the feature-arm regression weights do not stipulate a causal relationship - highly sophisticated causal relationships are difficult to formulate within LLMs due to the inherent difficulties of interpreting a neural network's internal layers; thus, in this paper, we focus on providing empirical studies and the conclusions we can draw from them. Finally, even with our rigorous studies to find the ROC-AUC Pareto-frontier, our reward model leverages LLM-as-judge, which may reflect the LLM's bias. Overall, these limitations posit potential directions by which the research community can further pursue and ultimately help expand our understanding of these powerful, albeit hallucinatory models.

## A.2 ETHICS & SOCIETAL IMPACT.

Our method alters inputs rather than model weights; it can reduce factually incorrect outputs but does not eliminate them. Failure modes include reward misspecification and domain shift. We report error analyses and release prompts to facilitate auditing and replication, as part of the appendix. Furthermore, we discuss the societal impact of hallucinations in the related works.

## A.3 REPRODUCIBILITY STATEMENT.

We aim to make our results fully reproducible. The main paper specifies the learning setup and algorithms (Algorithm 1; §3–§4), including the five rewrite arms with exact system-prompt templates (Table 9), the feature set used by the contextual policies (Table 11, Table 10), and the reward definition with its components and weights (§3, Table 6a, Figure 2). Evaluation datasets, splits, pre-processing, dataset-specific details, and licenses are detailed in §4 and Table 3; decoding/API configurations are documented here. For all experiments, we apply OpenAI's `gpt-4o-2024-08-06` with API parameters: temperature=0.2, top-p=1.0, frequency/presence penalties=0. We report statistical uncertainty (95% CIs) and paired significance tests, and provide ablations/sensitivity analyses through the paper that support our claims.

# B DISCUSSION ON RL AND BANDIT METHODS

**Remark 1** *Why bandits vs. full RL?* Within LLMs, for each input query, the transformer attends over the fixed context window and computes a softmax over the vocabulary to maximize token likelihood (Radford et al., 2019). Consequently, hallucinations occur at the moment of generation for that single query, making hallucination a **per-query** phenomenon (Huang et al., 2025a). Indeed, recent PPO variants for LLMs, such as GRPO (Shao et al., 2024) and ReMax (Li et al., 2024c), remove the critic via grouped Monte Carlo or baseline-adjusted returns, highlighting critic-free policies that our bandit formulations naturally generalize. Therefore, a full-episodic RL problem, which must solve a Markov decision process with long-horizon credit assignment and nonstationary transition dynamics (Sutton & Barto, 2018), can be practically suboptimal. Moreover, many of these methods rely on estimating a fixed average reward or state-action value $Q(s, a)$, which can obscure per-query idiosyncrasies; if the optimal rewrite arm varies sharply with linguistic context, a mere empirical average will yield suboptimal policies.

**Remark 2** *Link between Algorithm Choices and RL Methods.* Several algorithms we investigate in QueryBandits have analogues in RL: posterior sampling (PSRL) (Osband et al., 2013) as an analogue for Thompson sampling (Thompson, 1933); follow-the-regularized leader (FTRL) and its variants (Shalev-Shwartz et al., 2012), originating from proximal-gradient methods (Rockafellar, 1976) whose use in RL as proximal policy optimization (PPO) (Schulman et al., 2017) is well-established. Other PPO-style advances like DAPO (Yu et al., 2025) improve exploration-exploitation via dynamic sampling and reward filtering, and VAPO (Yue et al., 2025) demonstrates stable Long-

CoT training with an explicit value model–illustrating the spectrum from model-based to model-free approaches that contextual bandits sit within.

## C  TRUTHFULQA METRICS AND EVALUATION SETUP

TruthfulQA (Lin et al., 2022) offers several evaluation modes:

- ▸ **MC1 (single-true):** Given a multiple-choice question with four or five options, select the single true option. The model's choice is the option with the highest completion log probability; the score is accuracy over questions.
- ▸ **MC2 (multi-true):** Given a multiple-choice question with multiple reference answers labeled true or false, the score is the normalized total probability assigned to the set of true answers.
- ▸ **Generation:** Given a free-form question, generate a 1–2-sentence answer that maximizes truthfulness while maintaining informativeness. Metrics include GPT-judge and GPT-info (fine-tuned evaluators), BLEURT, ROUGE, and BLEU. A similarity-based score is computed as $\max_{\text{true}} \text{sim} - \max_{\text{false}} \text{sim}$.

In the main paper we focus on **MC1** for comparability across methods, as this regime aligns naturally with notions of *correctness* and *equivalence*. Zhang et al. (2024a) evaluate the **generation** setting using two fine-tuned GPT-3 classifiers (GPT-judge and GPT-info) to label responses for truthfulness and informativeness (binary classification). These labels are not accuracy and therefore are not directly comparable to our generative evaluation.

## D  SUMMARY OF BANDITS

- ▸ **Non-Contextual Adversarial**
  - **EXP3** (Auer et al., 2002b) Maintains weights $w_k$, samples $a_t \propto w_k$, updates $w_{a_t} \leftarrow w_{a_t} \exp\left(\frac{\gamma\, r_t}{K\, p_{a_t}}\right)$.
  - **FTPL** (Kalai & Vempala, 2005; Suggala & Netrapalli, 2020) Adds Gumbel noise $\xi_k \sim \text{Gumbel}(0, 1/\eta)$ (Gumbel, 1941) to cumulative rewards, selects $a_t = \arg\max(\text{cum\_reward}_k + \xi_k)$, then increments the chosen arm's reward.
- ▸ **Contextual Stochastic**
  - **LinUCB** (Lai & Robbins, 1985) Selects $a_t = \arg\max_k\left(x_t^\top \hat{\theta}_k + \alpha\sqrt{x_t^\top A_k^{-1} x_t}\right)$, updates $A_k \leftarrow A_k + x_t x_t^\top,\ b_k \leftarrow b_k + r_t x_t$.
  - **KL-UCB (LinUCB-KL)** (Garivier & Cappé, 2013) Replaces the UCB term with a KL-divergence-based confidence bound.
  - **Thompson Sampling** Maintains Gaussian posterior $\mathcal{N}(\mu_k, \Sigma_k)$; samples $\tilde{\theta}_k$, picks $a_t = \arg\max x_t^\top \tilde{\theta}_k$, updates the posterior.
- ▸ **Contextual Adversarial**
  - **FTRL** (McMahan, 2015) Selects arm maximizing $x_t^\top w_k - \lambda\|w_k\|_1$, with an $\ell_1$ regularizer.
  - $\epsilon$**-greedy FTRL** ...
  - **LinearEXP3** (Neu & Olkhovskaya, 2020) Contextual extension of EXP3, sampling arms based on exponentiated linear scores.
  - **LinearFTPL** (Hannan, 1957) Contextual adaptation of FTPL, applying Gumbel perturbations to linear reward estimates.

### D.1  LINUCB

The estimated parameter is:
$$\hat{\theta}_a = A_a^{-1}\mathbf{b}_a. \tag{2}$$

Given a query feature vector $\mathbf{x}$, the upper confidence bound (UCB) for arm $a$ is:
$$\text{UCB}_a(\mathbf{x}) = \mathbf{x}^\top \hat{\theta}_a + \alpha\sqrt{\mathbf{x}^\top A_a^{-1}\mathbf{x}}, \tag{3}$$

where $\alpha$ controls the exploration–exploitation trade-off. The arm selected is:
$$a^* = \arg\max_{a \in \mathcal{A}} \text{UCB}_a(\mathbf{x}). \tag{4}$$

---

**Algorithm 1** General Bandit + Rewrite Loop

---

**Require:** arms $\mathcal{A}$, context $x_t$, algorithm $\text{algo} \in \{$EXP3, FTPL, LinUCB, KL, FTRL, Thompson$\}$, hyperparameters
1: **for** $t = 1$ to $T$ **do**
2:     observe $x_t$
3:     **for** each arm $k \in \mathcal{A}$ **do**
4:         $s_k \leftarrow \text{Score}(\text{algo}, k, x_t)$
5:     **end for**
6:     select $a_t = \arg\max_{k \in \mathcal{A}} s_k$
7:     apply rewrite $a_t$ to query and observe reward $r_t$
8:     $\text{Update}(\text{algo}, a_t, x_t, r_t)$
9: **end for**

---

Table 3: **Datasets.** Overview of datasets, including domain, license, number of examples, associated scenarios, etc. These datasets span a diverse range of question types, domains, and reasoning skills, supporting robust evaluation. E = Extractive, M = Multiple Choice, A = Abstractive.

| Dataset | Scenario | Domain | License | Count | Citation |
|---|---|---|---|---|---|
| SQuADv2 | E, A | Wikipedia | CC BY-SA 4.0 | 86K | Rajpurkar et al. (2016; 2018) |
| TruthfulQA | M, A | General Knowledge | Apache-2.0 | 807 | Lin et al. (2022) |
| SciQ | M, A | Science | CC BY-NC 3.0 | 13K | Johannes Welbl (2017) |
| MMLU | M | Various | MIT | 15K | Hendrycks et al. (2021) |
| PIQA | M | Physical Commonsense | AFL-3.0 | 17K | Bisk et al. (2020) |
| BoolQ | M | Yes/No Questions | CC BY-SA 3.0 | 13K | Clark et al. (2019); Wang et al. (2019) |
| OpenBookQA | M | Science Reasoning | Apache-2.0 | 6K | Mihaylov et al. (2018) |
| MathQA | M | Mathematics | Apache-2.0 | 8K | Amini et al. (2019) |
| ARC-Easy | M | Science | CC BY-SA 4.0 | 5K | Clark et al. (2018) |
| ARC-Challenge | M | Science | CC BY-SA 4.0 | 2.6K | Clark et al. (2018) |
| WikiQA | A | Wikipedia QA | Other | 1.5K | Yang et al. (2015) |
| HotpotQA | A | Multi-hop Reasoning | CC BY-SA 4.0 | 72K | Yang et al. (2018) |
| TriviaQA | A | Trivia | Apache-2.0 | 88K | Joshi et al. (2017) |

Upon observing reward $r$, update:

$$A_a \leftarrow A_a + \mathbf{x}\mathbf{x}^\top, \quad \mathbf{b}_a \leftarrow \mathbf{b}_a + r\,\mathbf{x}. \tag{5}$$

## D.2 LINUCB+KL BANDIT STRATEGY

The algorithm is initialized with parameters: number of arms $n_{\text{arms}}$, dimension $d$, regularization parameter $\lambda$, exploration parameter $\alpha$, noise variance $\sigma_{\text{noise}}$, and KL-bound constant $c$. Each arm $a$ maintains a matrix $\mathbf{A}_a$ and a vector $\mathbf{b}_a$, initialized as $\lambda\mathbf{I}_d$ and $\mathbf{0}_d$, respectively.

The `select_arm` method computes the score for each arm $a$ using the following formulation:

$$\theta_a = \mathbf{A}_a^{-1}\mathbf{b}_a$$
$$\mu_a = \mathbf{x}^\top\theta_a$$
$$\text{var}_a = \mathbf{x}^\top\mathbf{A}_a^{-1}\mathbf{x}$$
$$n_a = \max(1, \text{counts}[a])$$
$$\text{raw\_bound}_a = \frac{\log(t) + c\log(\log(t+1))}{n_a}$$
$$\text{bound}_a = \max(\text{raw\_bound}_a, 0.0)$$
$$\text{bonus}_a = \sqrt{2 \cdot \text{var}_a \cdot \text{bound}_a}$$
$$\text{score}_a = \mu_a + \text{bonus}_a$$

where $\mathbf{x}$ is the context vector, $t$ is the time step, and $\text{counts}[a]$ is the number of times arm $a$ has been selected. The arm with the highest score is selected for exploration.

Table 4: **Instantaneous Regret.** Each cell reports mean per-step regret; **bold** marks the *minimum* per scenario. "Wins" counts per-family minima with ties split (0.5 each). "Macro-avg" is the unweighted average over scenarios. Static prompts sometimes win on regret by avoiding exploration, whereas contextual methods typically incur slightly higher immediate regret while delivering higher final accuracy (see Table 1), reflecting the exploration–exploitation tradeoff. NoRw = No-Rewrite

| Dataset | Base NoRw | Static Prompts | | | | | Non-Contextual | | | | Contextual Linear | | | | |
|---|---|---|---|---|---|---|---|---|---|---|---|---|---|---|---|
| | | Para | Simpl | Disamb | Clarify | Expand | EXP3 | FTPL | ε-FTRL | TS | LinUCB | LinUCB+KL | LinEXP3 | LinFTPL | TS |
| ARC-Challenge | **0.095** | 0.098 | 0.097 | 0.124 | 0.111 | 0.180 | 0.123 | 0.125 | 0.106 | 0.109 | 0.118 | 0.121 | 0.107 | 0.102 | 0.121 |
| ARC-Easy | 0.103 | 0.104 | 0.102 | 0.115 | 0.118 | 0.163 | 0.111 | 0.172 | 0.124 | **0.096** | 0.115 | 0.121 | 0.098 | 0.107 | 0.115 |
| BoolQA | 0.219 | 0.202 | 0.192 | 0.197 | 0.197 | 0.212 | 0.202 | **0.185** | 0.198 | 0.208 | 0.211 | 0.197 | 0.191 | 0.186 | 0.186 |
| HotpotQA | 0.198 | 0.203 | 0.199 | 0.191 | 0.206 | 0.201 | 0.197 | 0.199 | **0.188** | 0.197 | 0.191 | 0.197 | 0.192 | 0.196 | 0.194 |
| MathQA | **0.096** | 0.103 | 0.118 | 0.111 | 0.106 | 0.104 | 0.115 | 0.108 | 0.107 | 0.109 | 0.106 | 0.111 | 0.110 | 0.110 | 0.108 |
| MMLU | 0.134 | **0.130** | 0.153 | 0.142 | 0.150 | 0.168 | 0.139 | 0.143 | 0.146 | 0.143 | 0.139 | 0.145 | 0.144 | 0.168 | 0.139 |
| OpenBookQA | 0.160 | 0.159 | **0.157** | 0.218 | 0.228 | 0.341 | 0.223 | 0.169 | 0.177 | 0.159 | 0.200 | 0.198 | 0.243 | 0.221 | 0.188 |
| PIQA | 0.172 | 0.174 | 0.161 | 0.252 | 0.236 | 0.340 | 0.213 | 0.259 | 0.192 | 0.174 | 0.197 | 0.193 | 0.173 | **0.152** | 0.186 |
| SciQ (Abstract) | 0.147 | **0.135** | 0.158 | 0.153 | 0.155 | 0.179 | 0.150 | 0.176 | 0.149 | 0.137 | 0.156 | 0.155 | 0.174 | 0.176 | 0.150 |
| SciQ (MC) | 0.140 | **0.137** | 0.143 | 0.148 | 0.166 | 0.211 | 0.159 | 0.155 | 0.155 | **0.137** | 0.155 | 0.154 | 0.165 | 0.140 | 0.141 |
| SQuAD (Abstract) | 0.183 | **0.155** | 0.174 | 0.175 | 0.184 | 0.208 | 0.185 | 0.180 | 0.191 | 0.198 | 0.180 | 0.186 | 0.183 | 0.176 | 0.176 |
| SQuAD (Extract) | 0.139 | 0.129 | **0.128** | 0.165 | 0.169 | 0.244 | 0.166 | 0.130 | 0.148 | 0.168 | 0.165 | 0.154 | 0.147 | 0.133 | 0.141 |
| TriviaQA | **0.131** | 0.145 | 0.151 | 0.162 | 0.167 | 0.160 | 0.153 | 0.150 | 0.154 | 0.148 | 0.155 | 0.153 | 0.148 | 0.157 | 0.155 |
| TruthfulQA | 0.151 | 0.159 | 0.141 | 0.166 | 0.180 | 0.206 | 0.173 | 0.167 | **0.138** | 0.155 | 0.161 | 0.150 | 0.171 | 0.180 | 0.155 |
| TruthfulQA (MC) | 0.099 | 0.115 | **0.073** | 0.153 | 0.165 | 0.227 | 0.146 | 0.202 | 0.139 | **0.084** | 0.114 | 0.142 | 0.123 | 0.159 | 0.151 |
| WikiQA | 0.137 | 0.140 | 0.139 | 0.163 | 0.150 | 0.165 | 0.150 | 0.135 | 0.153 | **0.126** | 0.162 | 0.156 | 0.141 | 0.159 | 0.141 |
| Macro-avg | 0.144 | **0.140** | 0.148 | 0.160 | 0.163 | 0.216 | 0.166 | 0.159 | 0.157 | 0.155 | 0.160 | 0.159 | 0.160 | 0.156 | 0.160 |
| Wins | 3.0 | **3.5** | 3.0 | – | – | – | – | 1.0 | 2.0 | 2.5 | – | – | – | 1.0 | – |

| # Groups | Mean ROC–AUC | 95% CI |
|---|---|---|
| 5 | 0.9524 | [0.9165, 0.9884] |
| 10 | 0.9720 | [0.9549, 0.9891] |
| 15 | 0.9709 | [0.9581, 0.9836] |
| 25 | 0.9747 | [0.9674, 0.9821] |
| 50 | 0.9695 | [0.9633, 0.9756] |
| 75 | 0.9745 | [0.9688, 0.9801] |
| 100 | 0.9709 | [0.9626, 0.9792] |
| **150** | **0.9767** | **[0.9716, 0.9819]** |
| 200 | 0.9710 | [0.9653, 0.9767] |
| 300 | 0.9734 | [0.9709, 0.9758] |
| 400 | 0.9741 | [0.9713, 0.9769] |
| 500 | 0.9736 | [0.9703, 0.9769] |
| 600 | 0.9732 | [0.9701, 0.9763] |
| 700 | 0.9721 | [0.9695, 0.9748] |
| 800 | 0.9719 | [0.9699, 0.9738] |
| 900 | 0.9725 | [0.9716, 0.9734] |
| 1000 | 0.9737 | [0.9721, 0.9753] |
| **Macro-avg** | **0.9729** | – |

(a) Validity of the exploration-adjusted reward $r_{\text{adj}}$ as a correctness proxy. Mean ROC–AUC and 95% Confidence Intervals ($\pm 1.96$ SE); 10 resamples per $n$. By $\sim$150 groups, the CI lower bound exceeds 0.97.

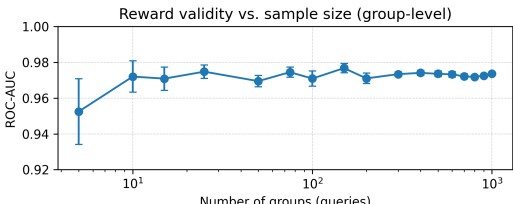

(b) Mean ROC-AUC vs. sample size $n$, with 95% CIs.

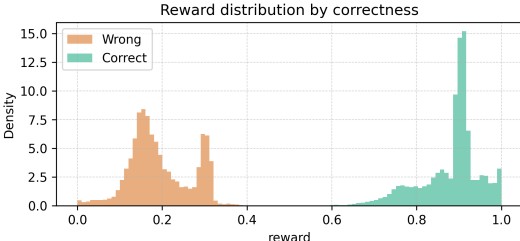

(c) Distribution of $r_t$ for correct vs. wrong (normalized density). Our reward presents a clear separation between our human validated labels. Per dataset reward distributions are located in Figure 14.

Figure 6: **Summary of reward validity. Left:** (a) numerical ROC–AUC and CIs across sample sizes. **Right:** (b) power curve; (c) class-conditional reward histogram of $r_t$ vs. human labels.

The `update` method updates the matrix $\mathbf{A}_a$ and vector $\mathbf{b}_a$ for the selected arm $a$ based on the received reward $r_t$:

$$\mathbf{A}_a \leftarrow \mathbf{A}_a + \mathbf{x}\mathbf{x}^\top$$
$$\mathbf{b}_a \leftarrow \mathbf{b}_a + r_t\mathbf{x}$$
$$\text{counts}[a] \leftarrow \text{counts}[a] + 1$$

This strategy leverages the KL-bound to dynamically adjust exploration bonuses, enhancing the LinUCB algorithm's ability to balance exploration and exploitation in a contextual setting.

Table 5: **TruthfulQA MC1 comparison.** $\Delta$ reports absolute percentage–point change vs our No–Rewrite baseline (80.7%). *QueryBandits* achieves the best score (LinUCB 88.8%, +8.1 pp) and strong Non–Contextual TS (88.7%, +8.0 pp); Contextual TS also improves (+4.5 pp). Closed GPT baselines cluster near ~81%, while open–model interventions reported on Llama–2–7B remain far below the GPT–4o baseline (e.g., TruthX 54.22%, $-26.5$ pp). Results across families highlight that context–aware linear bandits (LinUCB) are most effective on MC1, with TS (Non–Contextual) close but lacking per–query adaptation.

| Method | Backbone | MC1 (%) | $\Delta$ (pp) | Source | Notes |
|---|---|---|---|---|---|
| *QueryBandits (ours)* | | | | | |
| **Best (Dataset): LinUCB** | GPT–4o | **88.8** | **+8.1** | Closed | – |
| **Best (Overall): Contextual TS** | GPT–4o | 85.2 | +4.5 | Closed | – |
| **Best (Non–Contextual): TS** | GPT–4o | 88.7 | +8.0 | Closed | – |
| **Best Static: Simplify** | GPT–4o | 83.4 | +2.7 | Closed | No learning |
| **Worst Static: Expand** | GPT–4o | 67.9 | -12.8 | Closed | ——"—— |
| **No–Rewrite (Baseline)** | GPT–4o | 80.7 | 0.0 | Closed | Baseline for $\Delta$ |
| *Closed models (reference points)* | | | | | |
| **GPT–4o** | GPT–4o | 81.4 | +0.7 | Closed | OpenAI et al. (2024) |
| **GPT–4** | GPT–4 | 81.3 | +0.6 | Closed | ——"—— |
| **GPT–4o mini** | GPT–4o mini | 66.5 | -14.2 | Closed | ——"—— |
| **GPT–3.5 Turbo** | GPT–3.5 Turbo | 53.6 | -27.1 | Closed | ——"—— |
| *Open models: base / finetuned* | | | | | |
| **Llama–2–7B–Chat (base)** | Llama–2–7B–Chat | 34.64 | -46.1 | Open | Lin et al. (2022) |
| **Supervised Finetuning** | Llama–2–7B–Chat | 24.20 | -56.5 | Open | Zhang et al. (2024a) |
| *Contrastive decoding (open models)* | | | | | |
| **Contrastive Decoding (CD)** | Llama–2–7B–Chat | 24.40 | -56.3 | Open | Li et al. (2023) |
| **Decoding by Contrasting Layers (DoLa)** | Llama–2–7B–Chat | 32.20 | -48.5 | Open | Chuang et al. (2024) |
| **Self-Highlighted Hesitation (SH2)** | Llama–2–7B–Chat | 33.90 | -46.8 | Open | Kai et al. (2024) |
| **Induce-then-Contrast Decoding (ICD)** | Llama–2–7B–Chat | 46.32 | -34.4 | Open | Zhang et al. (2024b) |
| *Representation editing (open models)* | | | | | |
| **Contrast-Consistent Search (CCS)** | Llama–2–7B–Chat | 26.20 | -54.5 | Open | Burns et al. (2023) |
| **Inference Time Intervention (ITI)** | Llama–2–7B–Chat | 34.64 | -46.1 | Open | Li et al. (2024a) |
| **Truth Forest (TrFr)** | Llama–2–7B–Chat | 36.70 | -44.0 | Open | Chen et al. (2024b) |
| **TruthX** | Llama–2–7B–Chat | 54.22 | -26.5 | Open | Zhang et al. (2024a) |
| *Legacy references (TruthfulQA paper, MC)* | | | | | |
| **GPT–3 175B** | GPT–3 175B | 21.0 | -59.7 | Closed | Lin et al. (2022) |
| **GPT–J 6B** | GPT–J 6B | 20.0 | -60.7 | Open | ——"—— |
| **GPT–2 1.5B** | GPT–2 1.5B | 22.0 | -58.7 | Open | ——"—— |
| **UnifiedQA 3B** | UnifiedQA 3B | 19.0 | -61.7 | Open | ——"—— |

Table 6: **Inter-model agreement on the LLM-as-judge labels over 1,000 validation queries.** Values reported are fraction of exact label agreement, Cohen's $\kappa$, and Matthews correlation coefficient (MCC).

| Model A | Model B | % Agree | Cohen's $\kappa$ | MCC |
|---|---|---|---|---|
| gpt-5-2025-08-07 | gpt-5-mini-2025-08-07 | 0.960 | 0.916 | 0.916 |
| gpt-4.1-2025-04-14 | gpt-4o-2024-11-20 | 0.925 | 0.826 | 0.826 |
| gpt-4o-2024-11-20 | gpt-5-2025-08-07 | 0.909 | 0.802 | 0.810 |
| gpt-4.1-2025-04-14 | gpt-5-2025-08-07 | 0.906 | 0.794 | 0.807 |
| gpt-4o-2024-11-20 | gpt-5-mini-2025-08-07 | 0.903 | 0.790 | 0.801 |
| gpt-4.1-2025-04-14 | gpt-5-mini-2025-08-07 | 0.900 | 0.782 | 0.798 |
| gpt-5-mini-2025-08-07 | gpt-5-nano-2025-08-07 | 0.886 | 0.770 | 0.783 |
| gpt-5-2025-08-07 | gpt-5-nano-2025-08-07 | 0.882 | 0.762 | 0.778 |
| gpt-4o-2024-11-20 | gpt-5-nano-2025-08-07 | 0.823 | 0.642 | 0.680 |
| gpt-4.1-2025-04-14 | gpt-5-nano-2025-08-07 | 0.814 | 0.623 | 0.669 |

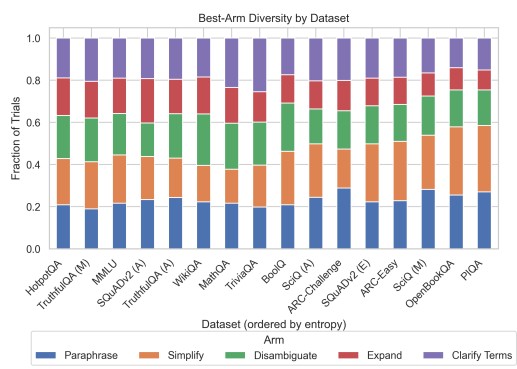

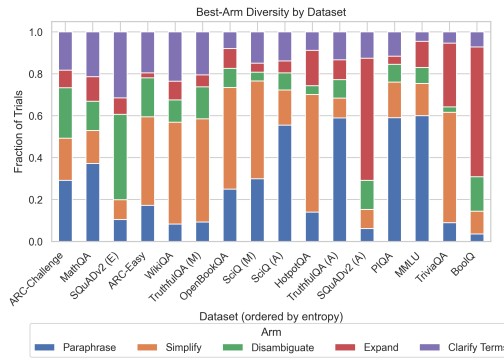

(a) Arm Diversity for Contextual Bandits, as a Fraction of Trials.

(b) Arm Diversity for Non-Contextual Bandits, as a Fraction of Trials.

Figure 7: For Non-Contextual bandits, *almost every* dataset is dominated by a single arm with the highest global reward (typically 40%-60% of the trials). The remaining 40-60% is split among the other four arms as noise, the non-contextual policy has no way to "know" when within a dataset a different arm might do better. In contrast, Contextual bandits show a more even mix: the top arm is only ∼25-30%, with two or three other arms contributing sizable shares (15-25% each). The contextual policy *reads the features* and diversifies its choices within each dataset.

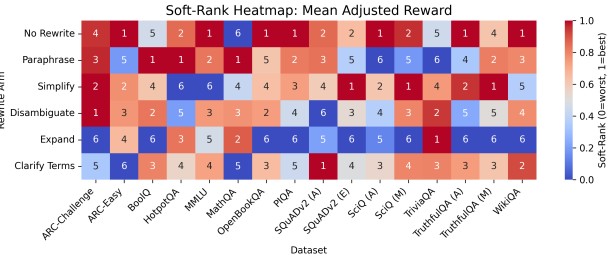

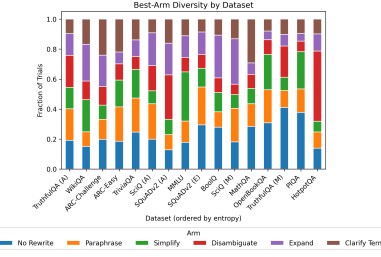

(a) Soft Rank Heatmap for all Bandits, including arm NO REWRITE.

(b) Arm Diversity when including NO REWRITE.

Figure 8: **Impact of the No-Rewrite Arm.** Note that these experiments are conducted on the original query "as-is" in the benchmark dataset, with no perturbations. Upon enabling the NO REWRITE option, our contextual bandit rapidly converges to this arm, which then achieves the highest reward on several datasets. We attribute this behavior to the LLM's tendency to memorize benchmark questions.

## D.3 FTRL

The algorithm is initialized with the following parameters: number of arms $n_{\text{arms}}$, dimension $d$, learning rate $\alpha$, exploration parameter $\beta$, and regularization parameters $l_1$ and $l_2$. The cumulative gradient vectors for each arm are stored in $\mathbf{z}_a$, initialized as zero vectors of dimension $d$.

The weight vector $\mathbf{w}_a$ for each arm $a$ is computed as:

$$
w_i = \begin{cases} -\frac{z_i - \text{sign}(z_i) \cdot l_1}{\frac{\beta + \sqrt{n_i}}{\alpha} + l_2} & \text{if } |z_i| > l_1 \\ 0 & \text{otherwise} \end{cases}
$$

where $z_i$ is the cumulative gradient for the $i$-th feature of arm $a$, and $n_i$ is the cumulative squared gradient for the $i$-th feature. The arm with the highest score, calculated as the dot product of the weight vector $w$ and the context vector, is selected:

$$
a_t = \arg\max_{a \in \{1, \dots, n_{\text{arms}}\}} \left( \sum_{i=1}^{d} w_i \cdot \mathbf{x}_i \right)
$$

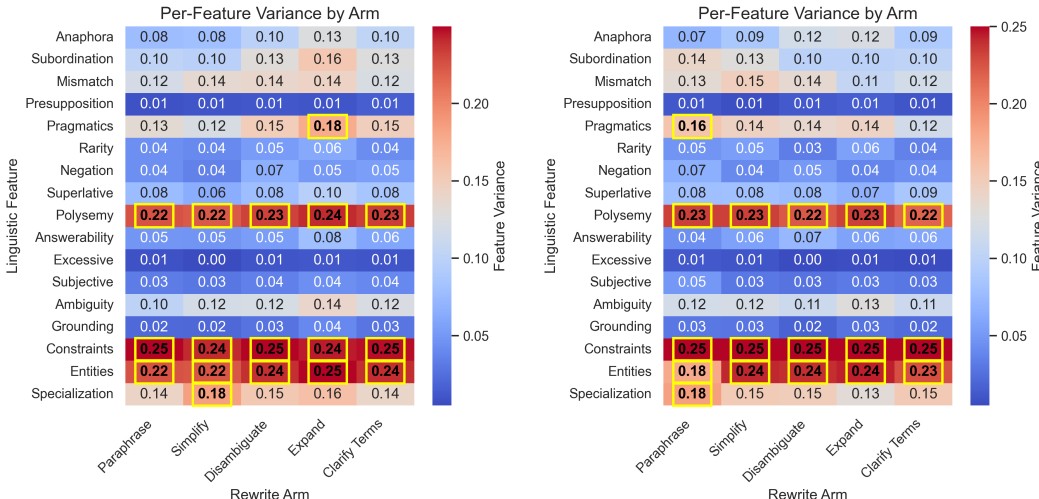

(a) Contextual Model Feature Variance.

(b) Non-Contextual Model Feature Variance.

Figure 9: Comparison of Feature Variance between (a) our contextual bandits and (b) its non-contextual counterparts. *Polysemy*, *Constraints* and *Entities* show the most variation. *Presupposition*, *Excessive Details*, and *Grounding* have the least.

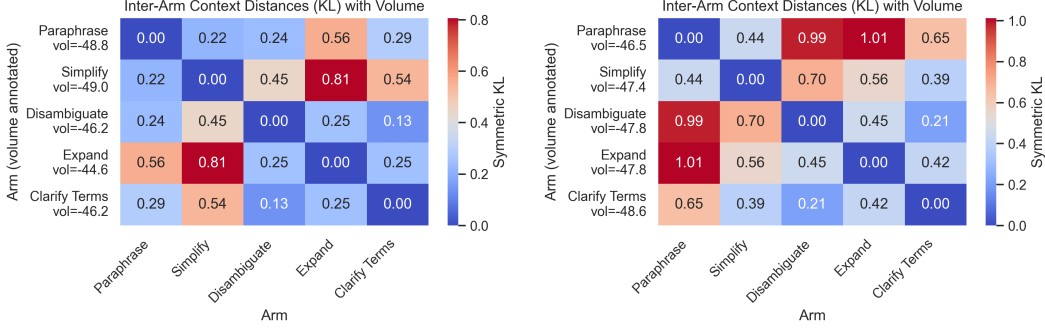

(a) Contextual Model KL Distance.

(b) Non-Contextual Model KL Distance.

Figure 10: Comparison of Inter-Arm Context Distances (Symmetric KL) between (a) our contextual bandits and (b) its non-contextual counterparts. Arm pairs such as EXPAND and PARAPHRASE in the non-contextual bandit setting exhibit high KL distances at 1.01. One interpretation is that the context-clouds barely overlap from dataset to dataset (Figure 7b).

Upon receiving a reward $r_t$ for the selected arm $a_t$, the algorithm updates the cumulative gradient vector $\mathbf{z}$ and the squared gradient sum $\mathbf{n}$ for the selected arm:

$$\varepsilon_{error} = \langle w, \mathbf{x} \rangle - r_t$$
$$g = \varepsilon_{error} \cdot \mathbf{x}$$
$$\sigma = \frac{\sqrt{n_i + g_i^2} - \sqrt{n_i}}{\alpha}$$
$$z_i \leftarrow z_i + g_i - \sigma \cdot w_i$$
$$n_i \leftarrow n_i + g_i^2$$

This formulation allows the FTRL algorithm to adaptively adjust the exploration-exploitation trade-off by incorporating both the cumulative reward and the uncertainty in the form of regularization terms, which are scaled by the learning rate $\alpha$ and exploration parameter $\beta$.

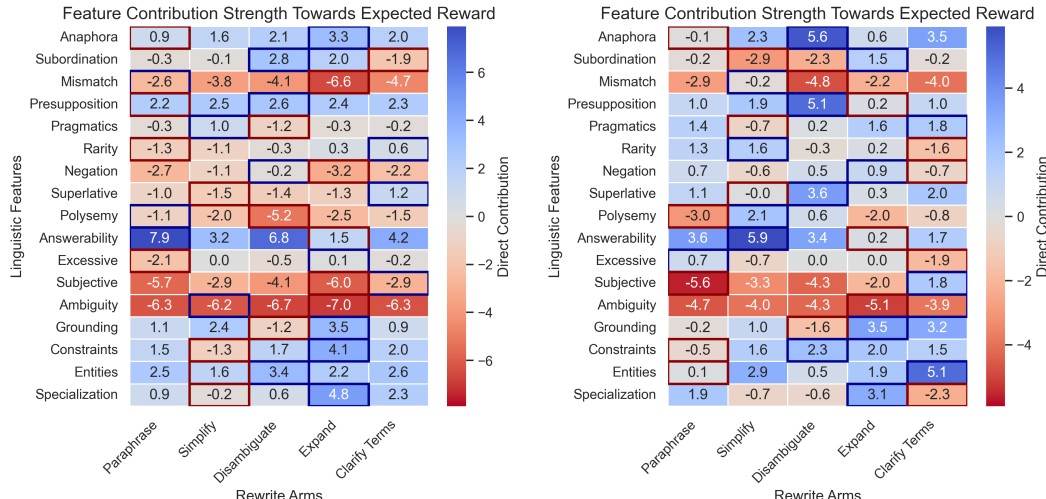

(a) Contextual Model Raw Feature Strength. (b) Non-Contextual Model Raw Feature Strength.

Figure 11: Comparison of Raw feature-level regression coefficients between (a) our contextual bandits and (b) its non-contextual counterparts. Each cell shows how enables a raw view into how specific linguistic feature changes the expected reward under each rewrite strategy.

## D.4 $\varepsilon$-Greedy Follow-The-Regularized-Leader (FTRL) Bandit Policy

At each round $t = 1, 2, \ldots, T$, we observe a contextual feature vector $x_t \in \mathbb{R}^d$ and must choose an arm $a_t \in \{1, \ldots, K\}$. For each arm $k$, the algorithm maintains a weight vector $w_{k,t} \in \mathbb{R}^d$ summarizing past feedback for that arm. We write

$$\mathcal{H}_{k,t-1} = \{(x_s, r_s) : s < t, a_s = k\}$$

for the history of rounds in which arm $k$ was selected, where $r_s \in [0, 1]$ is the observed reward. Given $x_t$ and the current weights $\{w_{k,t}\}_{k=1}^K$, FTRL defines a score for each arm via a linear model

$$\hat{r}_{k,t} = x_t^\top w_{k,t}.$$

We then apply an $\varepsilon$-greedy rule with exploration parameter $\varepsilon_t \in [0, 1]$:

▸ With probability $1 - \varepsilon_t$, choose the greedy arm

$$a_t = \arg \max_{k \in \{1, \ldots, K\}} \hat{r}_{k,t}.$$

▸ With probability $\varepsilon_t$, choose a uniformly random arm from $\{1, \ldots, K\}$.

In our experiments we use a fixed $\varepsilon$ ($\varepsilon = 0.10$), but standard decaying schedules such as $\varepsilon_t = \min\{1, c/\sqrt{t}\}$ are also compatible with the framework. After selecting $a_t$ and observing reward $r_t \in [0, 1]$, we update only the parameters associated with the chosen arm. Let

$$g_t = -r_t \, x_t$$

denote the (linear) loss gradient for arm $a_t$. FTRL defines the next iterate $w_{a_t, t+1}$ as the solution of a regularized cumulative optimization problem:

$$w_{a_t, t+1} = \arg \min_{w \in \mathbb{R}^d} \left\{ \sum_{s \le t : a_s = a_t} g_s^\top w + \lambda \Omega(w) \right\}, \tag{6}$$

where $\Omega$ is a convex regularizer and $\lambda > 0$ is a regularization coefficient. In our implementation we use an $\ell_2$-regularizer, $\Omega(w) = \frac{1}{2}\|w\|_2^2$, which yields a closed-form solution equivalent to online ridge regression over past rewards for that arm:

$$w_{a_t, t+1} = \left( \lambda I + \sum_{s \le t : a_s = a_t} x_s x_s^\top \right)^{-1} \left( \sum_{s \le t : a_s = a_t} r_s x_s \right).$$

Weights for all other arms $k \neq a_t$ remain unchanged, i.e., $w_{k,t+1} = w_{k,t}$. This $\varepsilon$-greedy FTRL variant thus behaves like a linear contextual bandit with a ridge-regularized FTRL learner for each arm, combined with a simple $\varepsilon$-greedy exploration mechanism. In practice, we do not recompute the closed-form solution from scratch; instead, we maintain sufficient statistics for each arm and update them incrementally.

## D.5 LINEAR EXP3

The algorithm is initialized with parameters: number of arms $n_{\text{arms}}$, dimension $d$, exploration parameter $\gamma$, and learning rate $\eta$. Each arm $a$ maintains a parameter vector $\theta_a$, initialized as $\mathbf{0}_d$.

We compute the probability distribution over arms using the following formulation:

$$\text{logits}_a = \theta_a^\top \mathbf{x}$$
$$\text{logits} = \text{logits} - \max(\text{logits})$$
$$\text{exp\_logits}_a = \exp(\text{logits}_a)$$
$$\text{base\_probs}_a = \frac{\text{exp\_logits}_a}{\sum_{a=1}^{n_{\text{arms}}} \text{exp\_logits}_a}$$
$$\text{probs}_a = (1 - \gamma) \cdot \text{base\_probs}_a + \frac{\gamma}{n_{\text{arms}}}$$

where $\mathbf{x}$ is the context vector. The arm is selected based on the probability distribution probs.

The `update` method updates the parameter vector $\theta_a$ for the selected arm $a$ using the estimated reward $\hat{r}_t$:

$$\hat{r}_t = \frac{r_t}{p_a}$$
$$\theta_a \leftarrow \theta_a + \eta \cdot \hat{r}_t \cdot \mathbf{x}$$

where $p_a$ is the probability of selecting arm $a$, and $r_t$ is the received reward. This strategy leverages exponential weighting and exploration bonuses to balance exploration and exploitation in a linear contextual setting.

## D.6 LINEAR FTPL

The algorithm is initialized with parameters: number of arms $n_{\text{arms}}$, dimension $d$, and learning rate $\eta$. Each arm $a$ maintains a parameter vector $\theta_a$, initialized as $\mathbf{0}_d$.

The `select_arm` method computes the perturbed scores for each arm using the following formulation:

$$\text{linear\_score}_a = \theta_a^\top \mathbf{x}$$
$$\text{noise}_a \sim \text{Gumbel}(0, \frac{1}{\eta})$$
$$\text{score}_a = \text{linear\_score}_a + \text{noise}_a$$

where $\mathbf{x}$ is the context vector. The arm with the highest perturbed score is selected:

$$a_t = \arg \max_{a \in \{1, \ldots, n_{\text{arms}}\}} \text{score}_a$$

$$\theta_a \leftarrow \theta_a + r_t \cdot \mathbf{x}$$

This strategy leverages random perturbations from a Gumbel distribution to balance exploration and exploitation, allowing the algorithm to explore suboptimal arms while exploiting the accumulated knowledge of their performance in a linear contextual setting.

| Stage | Median Tokens | Mean Tokens |
|---|---|---|
| Original query (input) | 16 | 19.3 |
| Feature-tagger output | 110 | 110.0 |
| Rewrite input | 26 | 29.3 |
| Rewrite output | 18 | 28.1 |
| Answer input | 64 | 91.3 |
| Answer output | 70 | 157.8 |
| Judge (input + output) | 162 | 252.3 |
| **Total** | **493** | **688** |

Table 7: **Token-level breakdown per query for QueryBandits.** The total corresponds to a per-query cost of approximately \$0.00035 at `gpt-4o-2024-11-20` pricing.

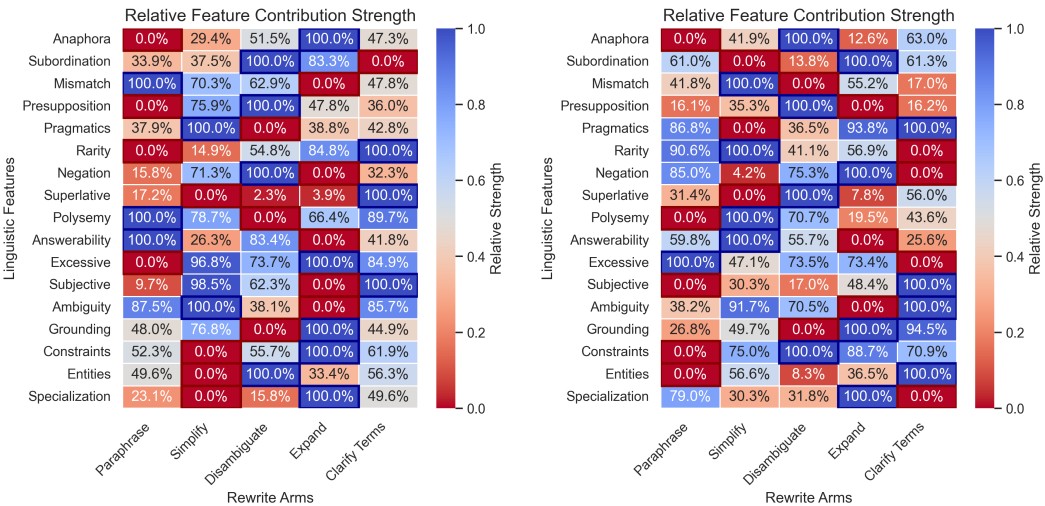

(a) Contextual Model Relative Feature Strength.  (b) Non-Contextual Model Relative Feature Strength.

Figure 12: Comparison of Min-Max Normalized feature-level regression coefficients between (a) our contextual bandits and (b) its non-contextual counterparts. Each cell shows how enables a relative view into how specific linguistic feature changes the expected reward under each rewrite strategy. Table 8 highlights contextual bandit trends.

## D.7 THOMPSON SAMPLING

For a given $\mathbf{x}$, sample $\tilde{\theta}a \sim \mathcal{N}(\mu_a, \Sigma_a)$ and select the arm maximizing:

$$a^* = \arg\max_{a \in \mathcal{A}} \mathbf{x}^\top \tilde{\theta}_a. \tag{7}$$

Standard Bayesian linear regression updates are then used to update $\mu_a$ and $\Sigma_a$ based on the observed reward $r$.

$$\Sigma_a^{-1} \leftarrow \Sigma_a^{-1} + \frac{1}{\sigma^2} \mathbf{x}\mathbf{x}^\top,$$
$$\mu_a \leftarrow \Sigma_a \left( \Sigma_a^{-1} \mu_a + \frac{1}{\sigma^2} \mathbf{x}\, r \right). \tag{8}$$

Table 8: **Top Drivers ($f^+_{\max}$) and Reducers ($f^-_{\max}$) of Rewrite Strategies per Linguistic Features** For each rewrite arm, we list the feature whose normalized coefficient was highest (100 %) and lowest (0 %), alongside a brief rationale for its positive or negative impact on downstream reward.

| Arm $a$ | $f^+_{\max}$ | Interpretation | $f^-_{\max}$ | Interpretation |
|---|---|---|---|---|
| DISAMBIGUATE | Subordination (100 %) | Long or nested clauses benefit from targeted disambiguation, which isolates and clarifies the core semantic relation. | Polysemy (0 %) | Highly polysemous terms lead disambiguation to pick the wrong sense, degrading downstream reward. |
| SIMPLIFY | Pragmatics (100 %) | Pragmatic cues (e.g. discourse markers, politeness) guide safe simplification without loss of meaning. | Superlative (0 %) | Stripping superlative constructions removes essential comparative context, hurting reward. |
| EXPAND | Constraints (100 %) | Queries already rich in constraints (time, location, numeric bounds) gain precision when expanded with further qualifiers. | Ambiguity (0 %) | Underspecified queries offer no detail to expand, so further addition of terms only introduces noise. |
| PARAPHRASE | Answerability (100 %) | Paraphrasing queries that are already answerable refreshes wording while preserving solvability, boosting LLM performance. | Presupposition (0 %) | Altering queries with strong presuppositions can break implied assumptions, reducing effective reward. |
| CLARIFY TERMS | Rarity (100 %) | Defining rare or domain-specific terms anchors the LLM's understanding of technical queries. | Subordination (0 %) | Clarifications in convoluted sentences can introduce further parsing difficulty, impeding reward. |

(a) Contextual Model Feature Uplift.

(b) Non-Contextual Model Feature Uplift.

Figure 13: **Reward Uplift by Contextual Feature and Strategy.** Feature Uplift measures how much the presence of a binary feature changes the expected reward for a given rewrite arm, formally $\Delta(f_i, a) = \mathbb{E}[r_t \mid \text{arm} = a, f_i = 1] - \mathbb{E}[r_t \mid \text{arm} = a, f_i = 0]$. (a) Under the **contextual** bandit, the strongest positive uplifts come from **Answerability** ($\approx$ +17 uniformly) and **Grounding** (+15–18), while **Ambiguity** ($\approx$ –15 to –18) and **Subjectivity** ($\approx$ –10 to –14) impose the largest hits across all arms. Mid-range features like **Presupposition** and **Constraints** deliver modest boosts ($\approx$ 5), and **Excessive Details** and **Anaphora** slightly hurt performance ($\approx$ –5 to –7). (b) The **non-contextual** bandit amplifies these trends: **Answerability** and **Grounding** remain the top drivers ($\approx$ +18–20), but **Ambiguity** becomes even more detrimental ($\approx$ –17 to –18), and **Mismatch** drops to nearly –16 under some arms. Notably, the non-contextual model shows a stronger negative effect for **Excessive Details** (up to –12) and **Entities** ($\approx$ –6) than the linear one, suggesting it more sharply penalizes noisy contexts. Together, these heatmaps reveal which linguistic signals each rewrite strategy leverages (or struggles with), and how context vs. context-blind policies weigh them differently.

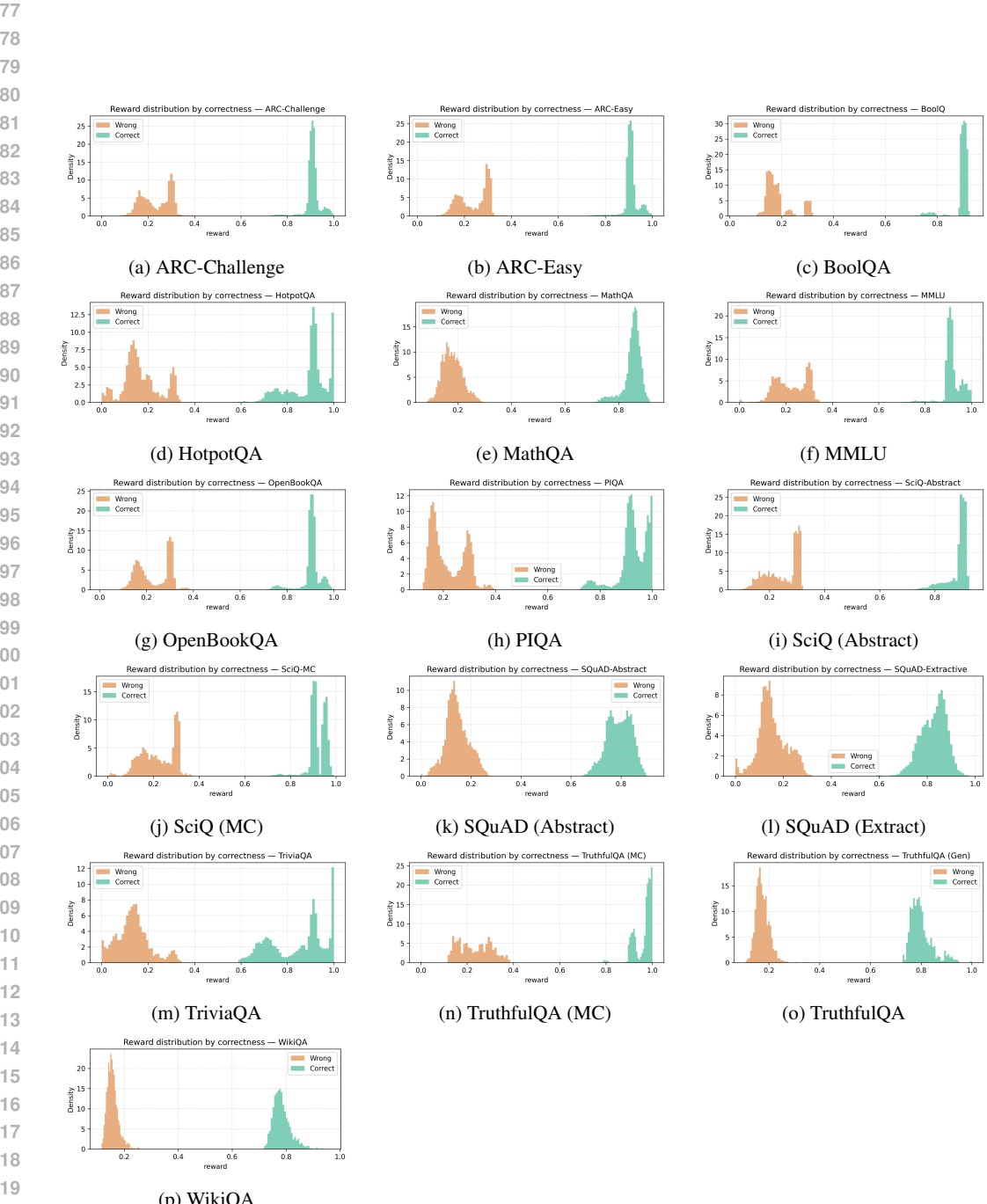

Figure 14: Per-dataset distributions of $r_t$ (normalized density).

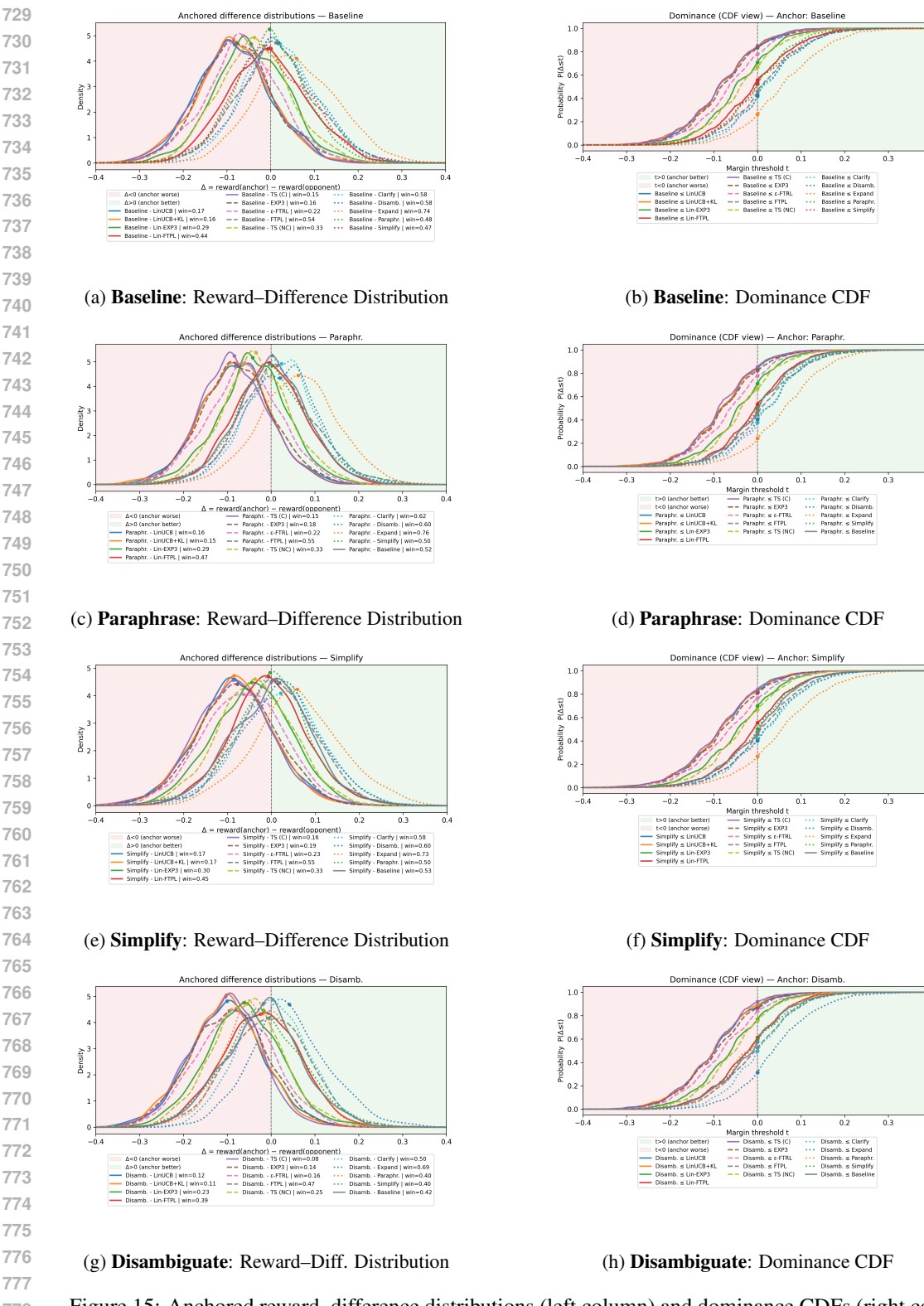

(a) **Baseline**: Reward–Difference Distribution

(b) **Baseline**: Dominance CDF

(c) **Paraphrase**: Reward–Difference Distribution

(d) **Paraphrase**: Dominance CDF

(e) **Simplify**: Reward–Difference Distribution

(f) **Simplify**: Dominance CDF

(g) **Disambiguate**: Reward–Diff. Distribution

(h) **Disambiguate**: Dominance CDF

Figure 15: Anchored reward–difference distributions (left column) and dominance CDFs (right column) for the baseline policy (**Baseline**) and core static rewrite strategies (**Paraphrase**, **Simplify**, **Disambiguate**). Each row fixes an anchor policy and compares its per-query reward against all competitors.

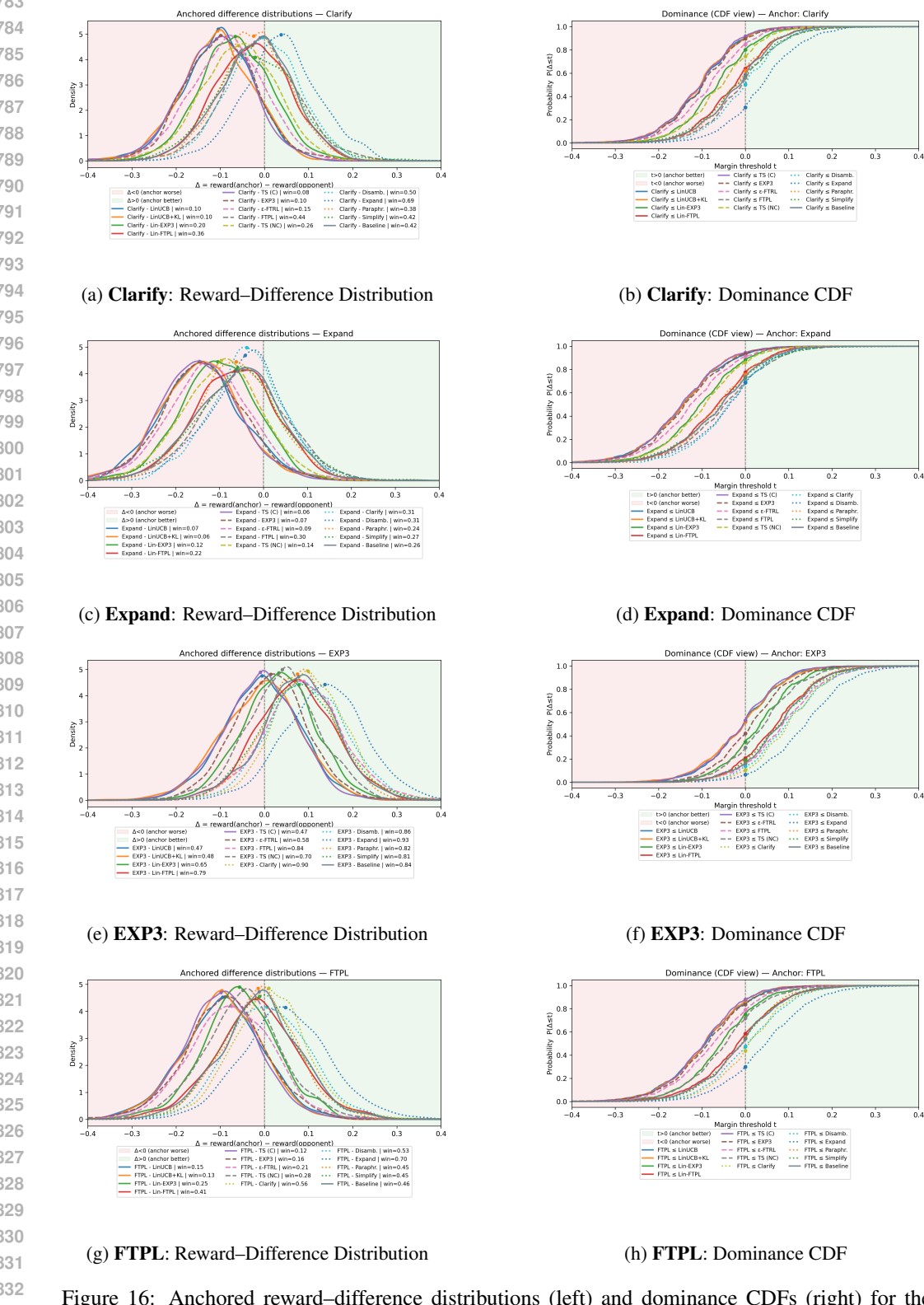

(a) **Clarify**: Reward–Difference Distribution

(b) **Clarify**: Dominance CDF

(c) **Expand**: Reward–Difference Distribution

(d) **Expand**: Dominance CDF

(e) **EXP3**: Reward–Difference Distribution

(f) **EXP3**: Dominance CDF

(g) **FTPL**: Reward–Difference Distribution

(h) **FTPL**: Dominance CDF

Figure 16: Anchored reward–difference distributions (left) and dominance CDFs (right) for the remaining static rewrite strategies (**Clarify**, **Expand**) and simple non-contextual bandits (**EXP3**, **FTPL**).

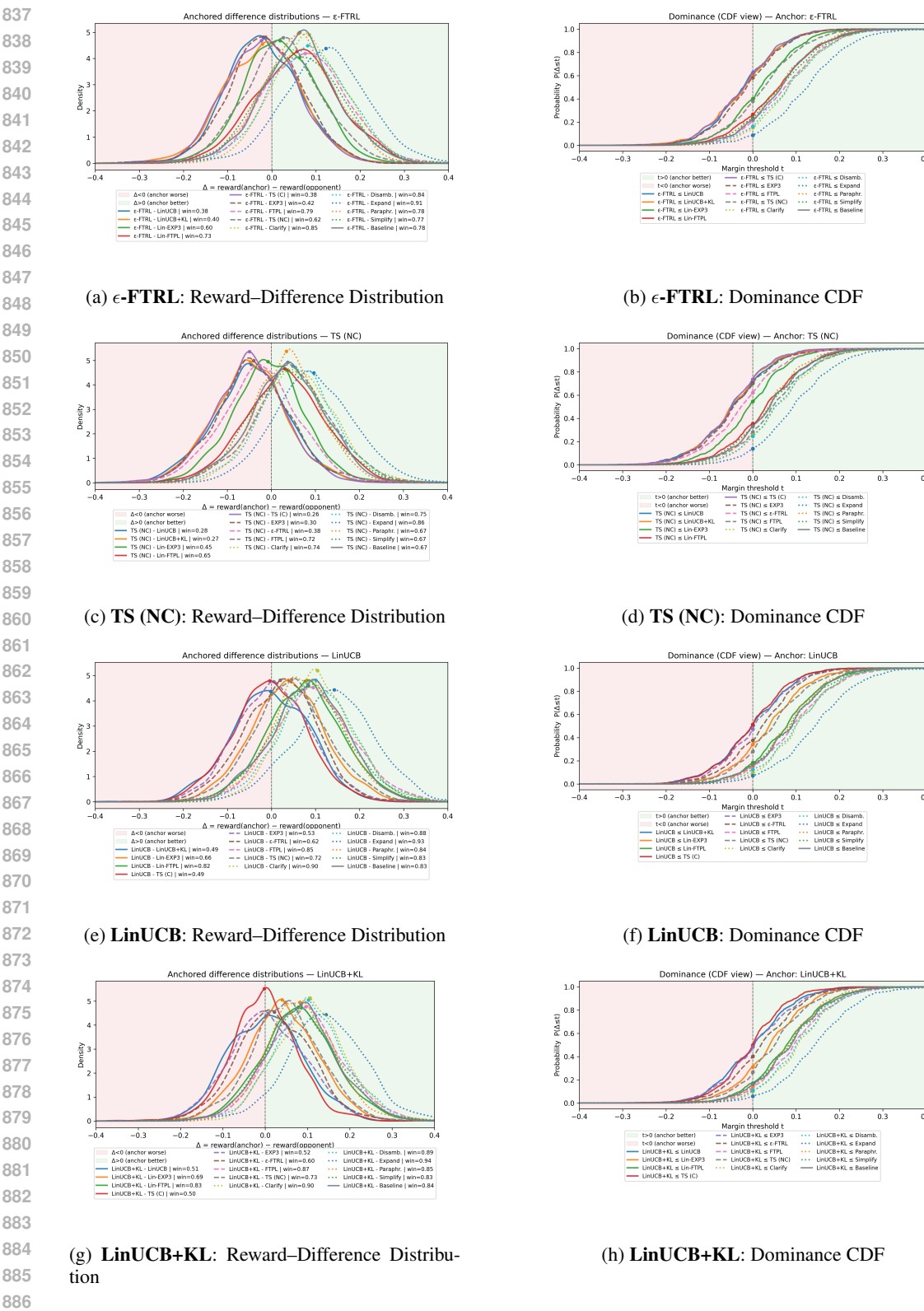

(a) $\epsilon$-**FTRL**: Reward–Difference Distribution

(b) $\epsilon$-**FTRL**: Dominance CDF

(c) **TS (NC)**: Reward–Difference Distribution

(d) **TS (NC)**: Dominance CDF

(e) **LinUCB**: Reward–Difference Distribution

(f) **LinUCB**: Dominance CDF

(g) **LinUCB+KL**: Reward–Difference Distribution

(h) **LinUCB+KL**: Dominance CDF

Figure 17: Anchored reward–difference distributions (left) and dominance CDFs (right) for advanced non-contextual bandits ($\epsilon$-**FTRL**, **TS (NC)**) and core contextual bandits (**LinUCB**, **LinUCB+KL**).

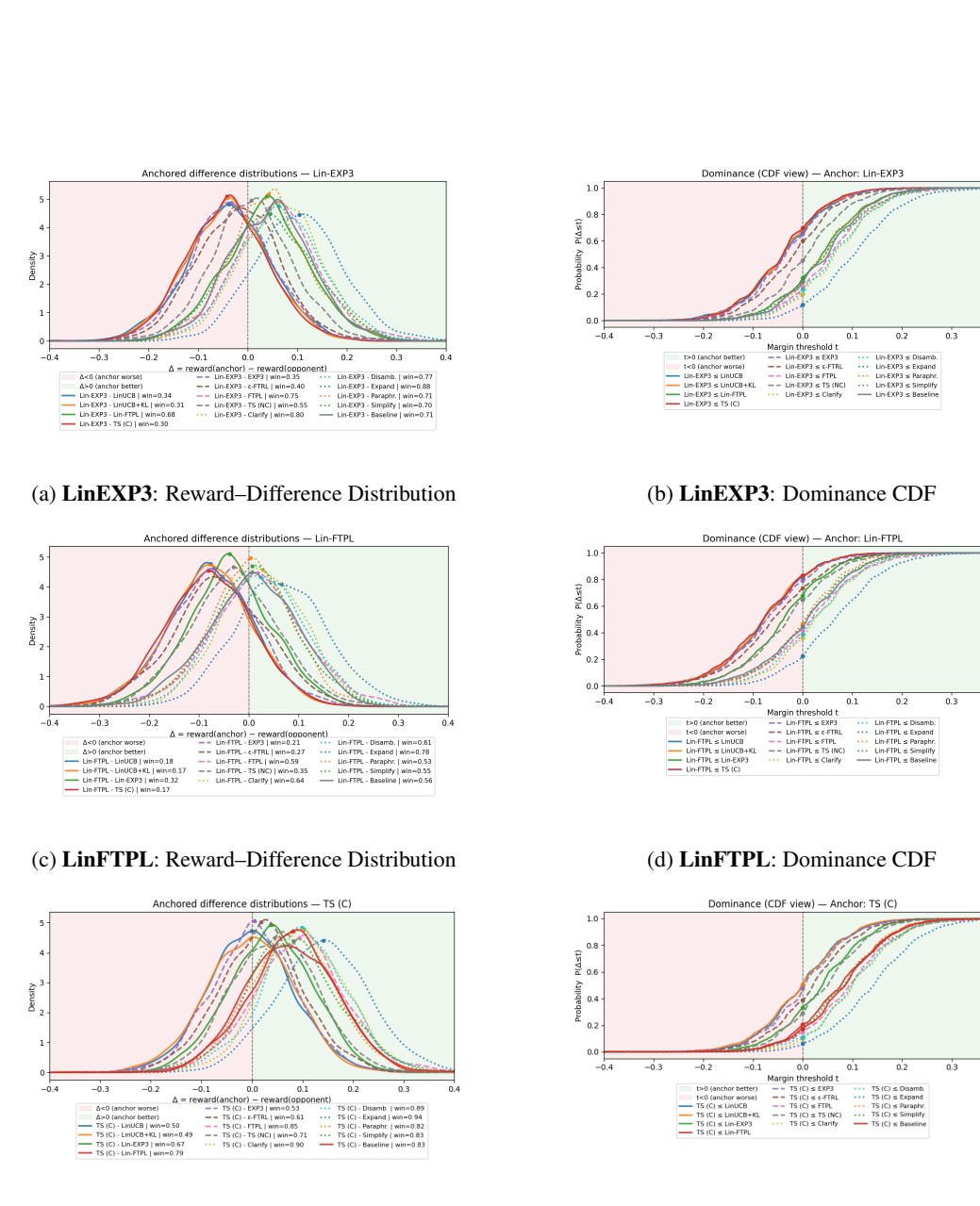

(a) **LinEXP3**: Reward–Difference Distribution

(b) **LinEXP3**: Dominance CDF

(c) **LinFTPL**: Reward–Difference Distribution

(d) **LinFTPL**: Dominance CDF

(e) **TS (Contextual)**: Reward–Difference Distribution

(f) **TS (Contextual)**: Dominance CDF

Figure 18: Anchored reward–difference distributions (left) and dominance CDFs (right) for the remaining contextual bandit policies (**LinEXP3**, **LinFTPL**, **TS (Contextual)**).

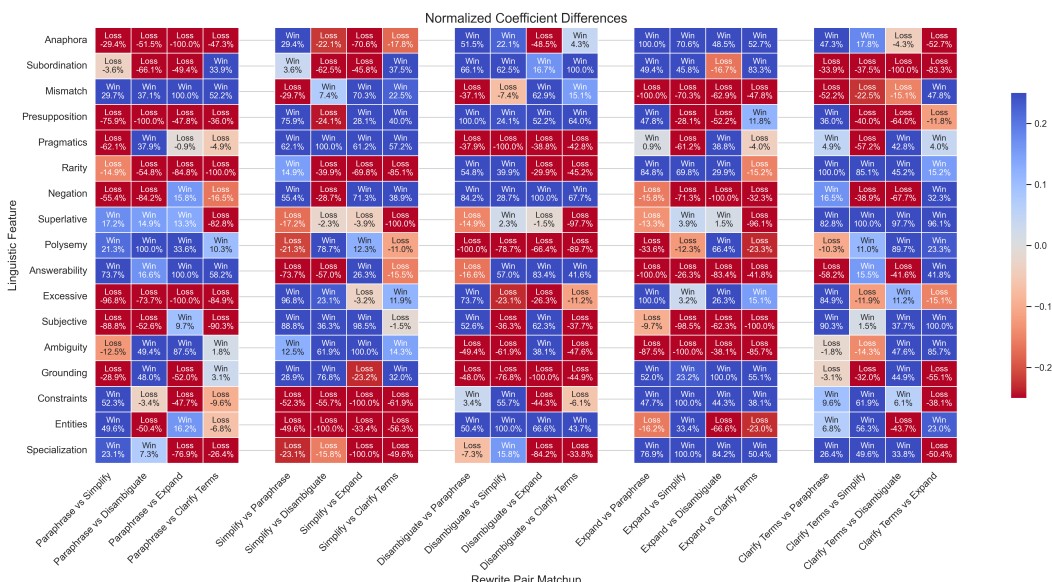

Figure 19: **Pairwise Normalized Coefficient Differences for Contextual Bandits.** Each cell shows the min–max–normalized difference in regression weight for a given linguistic feature (rows) between two rewrite arms (columns), e.g. "Paraphrase vs Disambiguate," "Simplify vs Expand," etc. Cells labeled "Win" (blue) indicate the feature favors the first arm in the matchup, while "Loss" (red) indicates it favors the second. Values are expressed as a percentage of the feature's full coefficient range.

Table 9: System prompt templates for rewrite arms. Replace {`original_query`} with the input at runtime. Each template must output *only* the rewritten query (no explanations).

| ID | Arm | System prompt template |
|---|---|---|
| $a_0$ | PARAPHRASE | You are a rewriting module. You will be given a user query: {`original_query`}. Rephrase it to improve clarity and introduce lexical diversity while strictly preserving semantic meaning, entities (including casing/accents), numbers, units, and constraints. Do not add or remove information. Output only the rewritten query. |
| $a_1$ | SIMPLIFY | You are a rewriting module. You will be given a user query: {`original_query`}. Simplify it by removing nested clauses and complex syntax. Use short, concrete phrasing (S–V–O order), keep all entities, numbers, units, and constraints, and avoid changing intent. Do not invent details. Output only the simplified query. |
| $a_2$ | DISAMBIGUATE | You are a rewriting module. You will be given a user query: {`original_query`}. Resolve vague references by replacing ambiguous pronouns (e.g., it/they/this) and temporal expressions with explicit, context-grounded referents and normalized dates. If a referent cannot be determined from the query alone, insert a bracketed placeholder (e.g., [ENTITY], [DATE]) rather than guessing. Preserve the original intent. Output only the disambiguated query. |
| $a_3$ | EXPAND | You are a rewriting module. You will be given a user query: {`original_query`}. Expand it by making implicit context explicit and adding salient, non-speculative attributes (e.g., scope, timeframe, location, units) that are entailed by the query. If crucial specifics are missing, insert neutral bracketed placeholders (e.g., [TIMEFRAME], [LOCATION]) instead of fabricating facts. Preserve the original intent and constraints. Output only the expanded query. |
| $a_4$ | CLARIFY TERMS | You are a rewriting module. You will be given a user query: {`original_query`}. Identify domain-specific jargon or terms of art and add concise parenthetical glosses (e.g., "term (brief definition)") where the meaning is standard and unambiguous. If uncertain, use a bracketed clarification placeholder (e.g., [DEFINE: TERM]) rather than guessing. Do not alter intent, entities, or constraints. Output only the clarified query. |

Table 10: Binary linguistic feature vector $\mathbf{f} \in \{0,1\}^{17}$ identified as challenging from a linguistics and LLM perspective. Features are grouped by type and grounded in prior work. For more specific examples, see Table 11.

| Feature | Description | Citation |
|---|---|---|
| *Structural Features* | | |
| Anaphora | Contains anaphoric references (e.g., *it*, *this*) | Schuster (1988); Chen et al. (2018) |
| Subordination | Contains multiple subordinate clauses (multi-clause structure) | Jeong et al. (2024); Blevins et al. (2023) |
| *Scenario-Based Features* | | |
| Mismatch | Question–task mismatch (e.g., open-ended query against retrieval-style task) | Gao et al. (2024); Kamath et al. (2024) |
| Presupposition | Assumptions within the query are implicitly regarded as truthful | Karttunen (2016); Levinson (1983) |
| Pragmatics | Requests phrased indirectly (e.g., *can you pass me the salt*) | Sravanthi et al. (2024); Levinson (1983) |
| *Lexical Features* | | |
| Rarity | Presence of rare words with poor representation | Schick & Schütze (2019); Khassanov et al. (2019) |
| Negation | Presence of negation (e.g., *not*, *never*) | Hossain & Blanco (2022); Truong et al. (2023) |
| Superlative | Presence of forms (e.g., *best*, *largest*) with implicit comparison sets | Pyatkin et al. (2024); Farkas & Kiss (2000) |
| Polysemy | Presence of words with multiple, related meanings | Ansell et al. (2021); Haber & Poesio (2024) |
| *Stylistic Complexity* | | |
| Answerability | Absence of speculative, sarcastic, or rhetorical phrasing | Qiao et al. (2023); Belfathi et al. (2023) |
| Excessive | Presence of excessive details/instructions that overload context; verbosity | Li et al. (2024b); Liu et al. (2023b) |
| Subjectivity | Query requires LLM to reflect creatively and engender a personal opinion | Durmus et al. (2024); Lv et al. (2024) |
| Ambiguity | Presence of ambiguous phrasing that opens multiple interpretations | Brown et al. (2020); Liu et al. (2023a) |
| *Semantic Grounding* | | |
| Grounding | Presence of a clear intent/goal statement | Clarke et al. (2009); Wei et al. (2023) |
| Constraints | Presence of temporal/spatial/task-specific constraints | Jiang et al. (2024); Lewis et al. (2021) |
| Entities | Presence of verifiable entities | Lee et al. (2023); Wang et al. (2023b) |
| Specialization | Query requires domain-specific knowledge for understanding | Watson et al. (2025b); Cho et al. (2024); Zeng et al. (2024) |

Table 11: Detailed Summary and Examples of Feature Categories, Definitions, and Examples (See Table 10). These definitions and examples become the prompts to create the binary context vector for our bandits.

| | Feature | Definition | Example |
|---|---|---|---|
| **Structural** | Anaphora | Presence of pronouns or references requiring external context. | "What about that one?" (Unclear reference) |
| | Subordination | Measures the presence of multiple subordinate clauses | "While I was walking home, I saw a cat that looked just like my friend's." |
| **Scenario-Based** | Mismatch | Mismatch between the query's intended output and its actual structure. | "Find me this paragraph in this document" (When document isn't given, this query cannot be answered) |
| | Presupposition | Unstated assumptions embedded in the query. | "Who is the musician that developed neural networks?" (Assumes such a musician exists) |
| | Pragmatics | Captures context-dependent meanings beyond literal interpretation. | "Can you pass the salt?" (A request, not a literal ability) |
| **Lexical** | Rarity | Use of rare or niche terminology. | "What are the ramifications of quantum decoherence?" (Uses low-frequency terms) |
| | Negation | Presence of negation words (*not*, *never*). | "Is it not possible to do this?" |
| | Superlatives | Detection of superlative expressions (*biggest*, *fastest*). | "What is the fastest algorithm?" |
| | Polysemy | Presence of ambiguous words with multiple related meanings. | "Explain how a bank operates." (Ambiguity: financial institution vs. riverbank) |
| **Stylistic** | Answerability | Assesses whether the query has a verifiable answer. | "What is the exact number of galaxies?" (Unanswerable) |
| | Excessive | Evaluates whether a query is overloaded with information, potentially distracting the model. | "Can you explain how convolutional neural networks work, including all mathematical formulas?" |
| | Subjectivity | Query requires the degree of opinion or personal bias | "What is the best programming language?" |
| | Ambiguity | Highly ambiguous context, task, and wording | "Tell me about history." (Too broad) |
| **Semantic** | Grounding | Evaluates how clearly the query's purpose is expressed. | "How does reinforcement learning optimize control in robotics?" (Clear intent) |
| | Constraints | Identifies explicit constraints (time, location, conditions) provided in the query. | "What was the inflation rate in the US in 2023?" |
| | Entities | Checks for the inclusion of verifiable named entities. | "Who founded OpenAI?" |
| | Specialization | Determines whether the query belongs to a specialized domain (e.g., finance, law). | "What are the legal implications of the GDPR ruling?" |

