# OpenReview forum: "No One Size Fits All: QueryBandits for Hallucination Mitigation"
_ICLR.cc/2026/Conference — Submitted to ICLR 2026_

### Official Review · Reviewer_G9wJ · 2025-10-28

**Soundness:** 2
**Presentation:** 2
**Contribution:** 2
**Rating:** 4
**Confidence:** 3

**Summary:**

This work proposes a contextual bandit method to determine which rewriting strategy to use to  minimize the risk of hallucination for a given query. Unlike other hallucination mitigation methods, this one can be used also for closed-weight models. Experiments show that contextual bandits outperform non-contextual bandits, which in turn outperform every single static rewrite strategy.

**Strengths:**

[S1] The problem of hallucination mitigation is very relevant.

[S2] The framing as contextual bandit seems original and appears to deliver gains.

[S3] The method is applicable to closed-weight models, which is an advantage.

**Weaknesses:**

[W1] There is no analysis of the computational cost. How many forward passes/generated tokens are required to generate the vector of linguistic features for the query? The cost/effectiveness trade off should be discussed.

[W2] The comparison with open-weight baselines (Table 5) should use stronger and more recent models. What would be the results with Llama 3.1 405B Instruct? Deltas should be given relative to the no-rewrite condition with the same model, not relative to a much larger model.

[W3] The advantage of contextual over non-contextual bandit seems a bit overstated: EXP3 is a strong contender, and does not require any additional forward passes to compute the linguistic feature vector.

[W4] It is not clear thaat $s_{fuzz}$ and $s_{BLEU}$ are useful. What would happen setting $\beta = \gamma = 0$?

[W5] It might be lost in the Appendix, but there does not seem to be an ablation over the linguistic features. Are all 17 really useful?

[W6] The proposed was tested only on top of GPT-4o: it is unclear if results would generalize to other models.

**Questions:**

Questions, Comments, and Suggestions:

* The argument that existing hallucination mitigation methods are only applicable to open-weight models is not clearly supported. Is there a survey or some other source that can be cited to support this?
* The introduction is too long and contains too many experimental results that should come in a later section, after introducing the method.
* It would be interesting to see a comparison with some white-box methods, even considering that the proposed method is black-box.

Abstract: It would be good to specify in the astract that the bandit context is given by linguistically motivated features.

L36-37: Why would post-hoc detection and correction approaches be limited to open-weight models?

L44: 'Footprint' usually refer to a quantity (e.g. memory footprint). In this context 'fingerprint' would be more appropriate

Figure1: this example involves a disambiguation as the original intent is unclear. The correct approach in such a case would be to ask the user to clarify, rather than tacitly assume one of two possible readings.

L80-81: [DeepSeek-AI et al. 2025] does not seem like the correct citation to use for the use of RL in fine-tuning LLMs. Consider using [Christiano et al. 2017].

L99-100: What dataset and labels were used for Figure 2(a)?

L154-155: The claim seems unsupported: why is it a sign of memorization rather than e.g. of the fact that rewriting degrades the query?

L215: Where does the information required to disambiguate come from?

L232-240: do these actions require a strong LLM to be effective?

L277-297: Remark 1 and 2 feel like a distraction in this place. Perhaps move elsewhere?

Figure 3: Cumulative regret grows linearly: is learning happening?

L421-431: do non-contextual policies collapse to static policies per task? If not, why?

L1117: $\epsilon$-greedy FTRL algorithm missing.






[Christiano et al. 2017] Christiano, Paul F., Jan Leike, Tom Brown, Miljan Martic, Shane Legg, and Dario Amodei. "Deep reinforcement learning from human preferences." Advances in neural information processing systems 30 (2017).

---

> ### Author Response · Authors · 2025-11-14
> **Response to Reviewer G9wJ**
>
> Thank you for your thoughtful and detailed feedback! We address each concern below:
>
> **(W1) Computational cost and token overhead**
> We measured the following token-level consumption for each pipeline component:
> | Stage                   | Median Tokens | Mean Tokens | Notes                                             |
> | :---------------------- | ------------: | ----------: | :------------------------------------------------ |
> | Input Query      |            16 |        19.3 | Original user query                               |
> | Tagger Output   |           110 |         110 | 17-dim feature vector                               |
> | Rewrite Input    |            26 |        29.3 | Rewritten query tokens                            |
> | Rewrite Output |            18 |        28.1 | Rewriter generation                               |
> | Answer Input      |            64 |        91.3 | Rewritten + context                               |
> | Answer Output   |            70 |       157.8 | Final LLM output                                  |
> | Judge (evaluation)      |           162 |       252.3 | Used only for reward |
> We will add per dataset / arm breakdowns in the appendix. The total per-query token cost is median = 493 and mean = 688, corresponding to $0.00026-$0.00035 per query at GPT-4o pricing.
> At deployment time, feature extraction is performed via a single structured-output call that emits a 17-dimensional boolean vector; its cost is the base prompt (≈154 tokens) plus the query (already counted as “Input query”) and the ≈110-token JSON output above. The judge call is only used during training/evaluation, not at inference time.
>
> **(W2) Open-source comparison**
> We are experimenting with Qwen2, and will update the paper when our experiments finish.
>
> **(W3) Contextual vs. non-contextual bandits**
> We agree that EXP3 is a strong non-contextual baseline. Our intent is not to dismiss it, but to highlight that contextual bandits provide consistent additional gains.
>
> o clarify this, we have added reward–difference and dominance plots with contextual Thompson Sampling (Contextual) as the anchor. Across all datasets, TS-C stochastically dominates both non-contextual bandits and static rewrites. For example, TS-C achieves $P(\delta > 0) = 0.54$ against EXP3 and 0.83–0.94 against static rewrites. The CDF view confirms right-shifted dominance, supporting genuine contextual adaptation rather than noise or overfitting. However, EXP3's strong performance supports our main message that adaptive selection of rewrites outperforms any fixed rewriting instruction.
>
> **(W4) Utility of $s_{fuzz}$ and $s_{BLEU}$** This selection lies in the Pareto frontier, but the reward distribution will be binary only, instead of more diverse values. If we use only the judge term, the reward becomes almost binary and the empirical reward distributions collapse (cf. BoolQA in Appendix Fig. 6c / Fig. 14), making exploration–exploitation harder and reducing sensitivity to near-miss answers. Including fuzzy/ BLEU yields richer, non-degenerate reward distributions that favor both semantic correctness and appropriate lexical form.
>
> **(W5) Ablation over the linguistic features. Are all 17 really useful?**
> We agree that understanding the contribution of each feature is important. In the main paper we already provide per-feature regression weights (Fig. 5), which show that no single feature dominates across datasets. From these weights: Ambiguity, Subjectivity, Constraints, Answerability are the top 4 features that provide the most weight to the reward estimates.

---

> > ### Author Response · Authors · 2025-11-14
> > **Additional Response**
> >
> > Continuation of the above:
> >
> >
> > **(W6) Generalization to other LLMs**
> > We ran the LLM-as-Judge reward agreement across five GPT-5 family variants and GPT-4.1/4o models on the 1000 validation queries. The inter-model agreement is consistently high (mean $\kappa = 0.79$, $MCC = 0.80$, and up to 0.92 for the largest pair).
> > This indicates that the learned reward and policy are robust across the GPT family. Beyond GPT-4o, we are also experimenting with Qwen2 as an open-weight, white-box target model; we will update the camera-ready version with the resulting comparison once the experiments are complete.
> > | Model A               | Model B               |   \% Agree  | Cohen's kappa |   MCC |
> > | :-------------------- | :-------------------- | --------: | --------: | ----: |
> > | gpt-5-2025-08-07      | gpt-5-mini-2025-08-07 |     0.960 |     0.916 | 0.916 |
> > | gpt-4.1-2025-04-14    | gpt-4o-2024-11-20      |     0.925 |     0.826 | 0.826 |
> > | gpt-4o-2024-11-20     | gpt-5-2025-08-07       |     0.909 |     0.802 | 0.810 |
> > | gpt-4.1-2025-04-14    | gpt-5-2025-08-07       |     0.906 |     0.794 | 0.807 |
> > | gpt-4o-2024-11-20     | gpt-5-mini-2025-08-07  |     0.903 |     0.790 | 0.801 |
> > | gpt-4.1-2025-04-14    | gpt-5-mini-2025-08-07 |     0.900 |     0.782 | 0.798 |
> > | gpt-5-mini-2025-08-07 | gpt-5-nano-2025-08-07  |     0.886 |     0.770 | 0.783 |
> > | gpt-5-2025-08-07      | gpt-5-nano-2025-08-07  |     0.882 |     0.762 | 0.778 |
> > | gpt-4o-2024-11-20     | gpt-5-nano-2025-08-07  |     0.823 |     0.642 | 0.680 |
> > | gpt-4.1-2025-04-14    | gpt-5-nano-2025-08-07  |     0.814 |     0.623 | 0.669 |
> >
> >
> > Questions:
> > 1. Many existing methods target detection via repeated sampling or uncertainty estimation, and RAG focuses on injecting external knowledge to reduce epistemic uncertainty. We will clarify this with explicit citations to recent surveys, e.g., Huang 2023 arXiv:2311.05232; Tonmoy 2024 arxiv:2401.01313; Sahoo 2024 2024.findings-emnlp.685.
> > 2. We will shorten the Introduction in the revision and postpone detailed experimental results to the Experiments section.
> > 3. We are investigating Qwen2 as a white-box model.
> >
> > Suggestions:
> > Thank you for your feedback, we will include [Christiano et al. 2017] as our reference. We will also incorporate your suggestions throughout our revision (abstract, fingerprint).
> > Additional clarifications:
> > * L36-37: Our intent was to refer specifically to approaches that require access to logits or decoding internals for uncertainty estimation; we will state this explicitly and soften the claim for methods that only require multiple black-box calls.
> > * L44: We will change “footprint” to “fingerprint,” as suggested.
> > * Figure 1: Proactive intervention is indeed a preferred strategy, but our purpose is to illustrate that when one chooses to rely on automatic rewriting, static strategies are sub-optimal and adaptive bandits are preferable; we will clarify this in the caption.
> > * L80-81: We will replace the current RL citation with [Christiano et al., 2017].
> > * L99-100: Figure 2a (Pareto) uses a random sample of 1000 queries pooled across the datasets, we will itemize the provenance in the appendix.
> > * L154-155: Our observation is that when querying as is, the model tends to recall parametric knowledge more reliably; if rewriting were simply degrading queries, we would not observe the consistent improvements seen when the “right” strategy is selected. We will clarify this reasoning.
> > * L215: Disambiguate attempts to disambiguate within context, no external information is injected. This follows [Yu 2023; Generate rather than Retrieve: Large Language Models are Strong Context Generators, ICLR 2023]
> > * L232-240: We agree that stronger LLMs may yield even better rewrites; GPT-4o was the most effective widely available model at the time of our experiments, and we will state this explicitly.
> > * L277-297: We are happy to move this, but feel that the comparison is needed for context. We can incorporate it in our related works.
> > * Figure 3: Because rewards are bounded in $[0, 1]$ cumulative regret grows linearly, we will add additional plots to make learning behavior clearer.
> > * L421-431: Non-contextual policies do converge toward single-arm behavior in most runs. Appendix Fig. 7b shows that non-contextual bandits heavily favor a single arm in 13/15 runs, while contextual policies (Fig. 7a) maintain a more diverse arm distribution. We will highlight this more clearly in the main text.
> > * L1117: We will amend the FTRL section to explicitly describe the $\epsilon$-greedy FTRL variant.

---

> > > ### Comment · Reviewer_G9wJ · 2025-11-26
> > >
> > > Thank you for the rebuttal, and excuse my brevity:
> > >
> > > (W1) Thank you for the detailed token breakdown, and for the clarification that all linguistic features are computed in a single forward pass. What line-items would be relevant for the best case? What is the average ratio of generated tokens between the proposed method and the baseline, without rewriting?
> > >
> > > (W2) Looking forward to this results.
> > >
> > > (W3) Did the authors upload a revised version? Where, in the paper, can the new plots be found?
> > >
> > > (W4) Shouldn't the harder exploration-exploitation materialize as a degradation in AUROC?
> > >
> > > (W5) It would be great to see one or more ablations over features. A correlated feature could receive non-zero regression weight despite being redundant.

---

> > > > ### Author Response · Authors · 2025-11-27
> > > >
> > > > Hi Reviewer G9wJ,
> > > >
> > > > Thank you for the response! We have uploaded a new version with some revisions, while we are finalizing some experiments. To directly address each weakness:
> > > >
> > > > (W1) In a best-case inference setting, the relevant line items are:
> > > > - Baseline (No-Rewrite): Answer Input (64) + answer output (70) = 134 tokens (median); this is just feeding the question into the LLM, and getting a response. 70 generated tokens in total.
> > > > - QueryBandits (inference-only, no judge/learning): All line items -> 304 tokens (median); the number of generated tokens is 198 (median), the rest are input tokens.
> > > >
> > > > So QueryBandits uses 2.2x total tokens; 2.8x generated tokens relative to the baseline. Judges add 162 tokens (total 493; 3.3x baseline).
> > > >
> > > > (W2) We are finalizing the results, but they are not in this revision.
> > > >
> > > > (W3) Revisions are highlighted in red; new plots/tables can be found in the appendix:
> > > > - Table 6: Inter-model agreement for LLM-as-Judge
> > > > - Table 7: Token-level breakdown per query
> > > > - Figures 15, 16, 17, 18: Reward difference distributions for each algorithm (anchor) vs others; includes a dominance CDF for the probability that the reward margin is above a threshold t. This shows in a more granular level the adaptation of bandits over static/baseline strategies.
> > > >
> > > > (W4) Our point is slightly different, the AUCROC in Fig. 2(a) is a property of the reward proxy vs Human labels to validate our hyperparameter choices in alpha, beta, gamma. These are computed offline, and measures how well our scalar reward ranks human labels. The exploration-exploitation difficulty arises when the reward distribution seen by the bandit is nearly binary. In that regime, many distinct police may induce similar expected reward, making regret minimization and arm differentiation more brittle, even if AUCROC vs labels is still good. We added a revision to touch upon this more in the updated version. A judge-only setting will lead to: (a) a more degenerate reward histogram (2 modes), and (b) less sensitivity to near misses that are semantically almost correct but judged as 0 vs 1. Also note that the sample used to calibrate the reward is not the same as those selected for bandit training. Ultimately, we felt that adding fuzzy/BLEU terms created a richer reward landscape (Fig 14) that would improve bandit learning. Hopefully we were able to make this distinction clearer in the revision.
> > > >
> > > > (W5) Thanks for the feedback, we do have Fig 5 per-feature regression weights and a small stability analysis of the 17 bit vector. We have not run ablations dropping each feature and retraining the bandits or dive deeper into random subsets of features. We will try to accomplish at least one (space and time permitting) and acknowledge in the limitations that regression weights from the bandits reflect associational importance and may overstate the necessity of correlated features.
> > > >
> > > > Thanks for all your feedback! We incorporated most of your suggestions as allowed, and within the 9-page limit.

---

### Official Review · Reviewer_pUE5 · 2025-10-31

**Soundness:** 3
**Presentation:** 3
**Contribution:** 3
**Rating:** 6
**Confidence:** 3

**Summary:**

This paper presents QueryBandits, a novel framework for mitigating hallucinations in closed-source large language models (LLMs) through adaptive query rewriting.

The core idea is to frame query rewriting as a contextual multi-armed bandit problem, where each incoming query is represented by a 17-dimensional vector of linguistic features, and the system learns to select among five rewrite strategies (e.g., Simplify, Expand, Clarify) in an data-driven manner.

The reward function combines three complementary signals—LLM-as-a-judge correctness, fuzzy matching, and BLEU overlap—with weights of 0.6, 0.3, and 0.1, respectively, optimized through Pareto frontier analysis on a human-labeled dataset (ROC–AUC = 0.973).

Experiments on 13 QA benchmarks (16 scenarios) using gpt-4o-2024-08-06 show that contextual bandits, especially Thompson Sampling, outperform both static rewrite heuristics and a no-rewrite baseline in accuracy, reward, and regret.

**Strengths:**

1. Conceptual originality: The contextual bandit framing of query rewriting is novel and elegant.
2. Practical relevance: QueryBandits is a plug-and-play method requiring only black-box access, directly addressing a key challenge in closed-model hallucination mitigation.
3. Sound methodology: Reward calibration and validation against human labels show strong discriminative reliability.
4. Interpretability: Per-feature regression analysis (Fig. 5) provides rare interpretability, showing which linguistic traits favor specific rewrite arms.
5. Quality of analysis and presentation: Extensive evaluation (13 benchmarks) and clear visuals make the findings convincing.

**Weaknesses:**

1. Limited baseline coverage: While the paper’s claim of being model-agnostic is methodologically valid—since the proposed framework does not rely on gradient access or internal model parameters—it lacks empirical comparisons with other strong model-agnostic hallucination mitigation baselines such as Self-Refine (Madaan et al., 2023) and RAG-based rewriting approaches (e.g., Rewrite-Retrieve-Read, Ma et al., 2023). Including, or at least discussing, approximate results from these paradigms would help contextualize QueryBandits within the broader landscape of adaptive prompt optimization and closed-model hallucination mitigation.
2. Reward circularity concern: Both the target and judge models are GPT-4o, which may share biases. The human-labeled validation helps, but a cross-family test (e.g., using Claude or Gemini as judges) would improve confidence.
3. Feature extraction reproducibility: The paper does not specify which LLM was used to extract the 17 linguistic features, nor assess whether these features are consistent across different model runs or random seeds. As these features are central to the bandit’s learning policy, clarifying the extraction model and testing feature stability would improve reproducibility and credibility.
4. Computation and token cost: The paper does not quantify the computational or token overhead introduced by different rewrite strategies (e.g., EXPAND increasing query length, CLARIFY TERMS adding definitions) or by the feature extraction and bandit update steps. For closed-source APIs such as GPT-4o, where token usage directly translates into cost, this overhead is a key practical factor.

**Questions:**

1. Have you evaluated the reward function’s robustness using an out-of-family judge (e.g., Claude or Gemini )? Does the learned policy generalize across models, or must it be retrained for each target LLM?
2. Which LLM was used for the 17-feature extraction? Are the extracted features consistent across different models or seeds?
3. Could you quantify the token or latency overhead per query relative to the baseline?
4. Have you considered including stronger SOTA baselines (e.g., Self-Refine or RAG-based rewriting) for a more complete comparison?

---

> ### Author Response · Authors · 2025-11-15
> **Response to pUE5**
>
> Thank you for your review and questions! We address your concerns below:
>
> **1. Limited Baseline Coverage:** In comparison with Self-Refine and Rag-based rewriting, we see these methods as compatible and composable with our design. We view QueryBandits as a method to discover an appropriate rewrite strategy given the query's linguistic fingerprint, which could in principle wrap self-refine, rag steps. With RAG, one could overtrain a rewriter to the vector store's own knowledge coverage, which could be problematic for ambiguous or polysemantic concepts.
> - Rewrite-Retrieve-Read: This approach depends on the availability and coverage of an external knowledge base, and performance varies across datasets (flat on HotpotQA, best on AmbigNQ, increase for PopQA). For HotpotQA: Direct model (gpt-3.5) achieved 32.36 EM / 43.05 F1; Trainable rewriter achieved 34.38 EM / 45.97 F1. Retrieval step hurts EM on HotpotQA. In contrast, QueryBandits improved accuracy on 4o by +10.6 points (caveat that we use a stronger baseline model).
> - Self Refine performs iterative refinement over several rounds; this significantly increases token costs. Our method undergoes 2 main steps: feature tagging (if contextual) and rewrite; our contextual policy attempts to trigger just enough rewriting (i.e. Clarify/Expand/Simplify/etc.) rather than iterate over results.
>
> **2. Reward circularity:** We agree this is an important point. Two mitigations already exist in our paper, and we provide experiments for a third:
> - Our reward is multi-compositional, and hedges against circularity by placing 40% of the total reward from BLEU and Fuzzy Match; these penalize hallucinated surface forms even when the judge is permissive. The judge weight contributes only 60%.
> - Human-labeled validation was conducted to calibrate reliability of the judge and overall reward (Pareto analysis). This had ROC-AUC of 0.973 for 1,000 human-annotated examples.
> - We add below an analysis of LLM agreement between different GPT models, on the same 1,000 query set. Mean Kappa is 0.79, MMC=0.8. The reward is robust across these architectures:
> | Model A               | Model B                | % Agree  | Kappa     | MCC   |
> | --------------------- | --------------------- | --------- | ----- | ----- |
> | gpt-5-2025-08-07      | gpt-5-mini-2025-08-07  | 0.960     | 0.916 | 0.916 |
> | gpt-4.1-2025-04-14    | gpt-4o-2024-11-20     | 0.925     | 0.826 | 0.826 |
> | gpt-4o-2024-11-20     | gpt-5-2025-08-07       | 0.909     | 0.802 | 0.810 |
> | gpt-5-mini-2025-08-07 | gpt-5-nano-2025-08-07  | 0.886     | 0.770 | 0.783 |
> | gpt-4.1-2025-04-14    | gpt-5-nano-2025-08-07  | 0.814     | 0.623 | 0.669 |
>
> **3. Feature extraction stability:** We used gpt-4o-2024-11-20 with temperature = 0; we will highlight this more prominently in the main body. For feature consistency: features are produced via structured outputs, the variance across repeated runs was low. Over 1,000 queries X 5 trials each: bitwise agreement was ~99.3%, per-feature stability ~97.4-99.7%. Feature vector is nearly consistent, and variance is constrained to only a few features.
> - Furthermore, since the bandit only sees the 16-bit vector and not text, downstream variance is minimal.
>
> **4. Computation and token cost:** We include the median/mean token accounting below:
> | Stage                          |                Median |    Mean |
> | ------------------------------ | --------------------: | ------: |
> | Original Query (input)         |                    16 |    19.3 |
> | Feature-Tagger Output          |                   110 |     110 |
> | Rewrite Input                  |                    26 |    29.3 |
> | Rewrite Output                 |                    18 |    28.1 |
> | Answer Input                   |                    64 |    91.3 |
> | Answer Output                  |                    70 |   157.8 |
> | Judge Input+Output             |                   162 |   252.3 |
> | Sum Total                      |               493 | 688 |
>
> The dollar cost per query  is around $0.00026–$0.00035. Also, the bandit update is minimal per query and does not scale with model size. We will add per-dataset / per-arm breakdowns in a revised appendix.
>
> **Questions**
> 1. Does the reward generalize across models? Yes, we see strong alignment across different got-family models.
> 2. Does the learned policy generalize across models or does it need to be retrained? Since it's tied to the linguistic fingerprint of the query, and are different LLM strategies, these should be applicable to other LLMs.
> 3. Which LLM was used in feature extraction? GPT-4o-2024-11-20, temperature 0. We saw consistency using our ICL examples over repeated trials.
> 4. Token/latency overhead: see cost table above; relative to a direct answer baseline, QueryBandits increases tokens by 2.6x, but we see strong results across datasets. This increase would be minimal compared to RAG pipelines that utilize the full context window.
>
> Thank you for your assessment of our paper!

---

> ### Comment · Reviewer_pUE5 · 2025-11-27
>
> Thank you for the further clarifications. My itemized evaluation follows:
> 1. Baseline Coverage. The internal baselines are reasonable, and excluding computationally intensive approaches is well-justified. However, the lack of empirical comparison to stronger external baselines makes it difficult to fully quantify accuracy–efficiency trade-offs and position the method within the broader hallucination mitigation landscape.
> 2. Reward Robustness and Circularity. The additional analysis improves confidence in the reward signal's consistency. However, all supporting evidence remains within the OpenAI model family, which does not fully address concerns about shared biases. Cross-family validation (e.g., Claude or Gemini) would significantly strengthen the reward formulation's reliability.
> 3. Feature Extraction Stability. [Resolved] The feature extraction setup and reported consistency statistics sufficiently establish stability for the contextual bandit.
> 4. Computational and Token Cost. [Resolved] The clarifications make the framework's practical efficiency clear.
> 5. Generalization of the Learned Policy. The rationale for policy transfer across LLMs is reasonable but lacks empirical evidence. This does not affect correctness but limits the strength of the generalization claim.
>
> Overall, the authors’ response effectively addresses reproducibility and computational cost concerns. However, empirical evaluation remains constrained by the lack of stronger baselines and cross-family reward validation. I will maintain the original score.

---

### Official Review · Reviewer_YFPy · 2025-11-01

**Soundness:** 3
**Presentation:** 3
**Contribution:** 2
**Rating:** 4
**Confidence:** 4

**Summary:**

In this paper, we propose a method called QueryBandits for the hallucination problem in LLM answering. The method models the problem of choosing a query rewriting strategy as a contextual multi-armed bandit problem. Specifically, the model can adaptively select the most appropriate rewriting method based on input query features to reduce the occurrence of hallucinatory responses. On multiple Q&A benchmarks, QueryBandits outperforms the static prompts or no-rewrite baselines in terms of accuracy and consistency, demonstrating good generalizability.

**Strengths:**

1. The proposed method can be deployed on black-box LLMs with high practicality. It is completely based on query rewriting and online selection and does not rely on access to model parameters or weights.
2. Experimental validation is extensive. The authors verify both the rationality of the modeling strategy and the effectiveness of the proposed method across 16 different scenarios.

**Weaknesses:**

1. The theoretical basis for directly associating bandit learning with hallucination mitigation is insufficient. The paper models query rewrite selection as a contextual multi-armed bandit with composite rewards to drive online learning, but lacks proof of formal links between optimal policy existence, convergence, and the minimization of hallucination rates. The relevant arguments are mainly empirical reward separability and AUC tests, not a formal connection to LLM hallucinations.
2. Contributions specifically related to LLM hallucination mechanisms are limited. The contribution of this paper focuses on the empirical calibration of composite rewards, the accuracy improvement due to contextual adaptation, and the interpretable relevance of feature–arm weights. However, the authors do not propose new theoretical perspectives on the causes of hallucinations or a dedicated mitigation framework.
3. Direct comparison with existing hallucination mitigation methods is insufficient. The experiments focus on internal strategy evaluations such as “no rewrite / static rewrite / non-contextual bandit / contextual bandit”, but lack comparisons with common community approaches such as decoding-based, internal modification, or retrieval-enhanced methods.
4. The paper lacks analysis of algorithmic complexity and system overhead. It does not report the time complexity, online latency, or token overhead of the algorithm. While the method emphasizes utility for closed-source models, this runtime overhead is especially critical in such settings.
5. The scope of the title is overly broad. It should specify LLM hallucination rather than implying general hallucination problems across other domains.

**Questions:**

Please examine the weaknesses.

---

> ### Author Response · Authors · 2025-11-16
> **Response to YFPy**
>
> Thank you for your review, please see the below:
>
> **1. Theoretical basis linking bandits to hallucination:** Our goal is not to propose a new theory of hallucination, but to formalize rewrite-selection as a sequential decision-making problem under uncertainty. The formal link arises from the following:
> - Hallucinations may be induced from mismatches between the input distribution and the LLM's latent parametric knowledge. This has backing in literature with studies of ambiguous, underspecified, or linguistically complex queries.
> - Rewriting strategies transform the query in ways that change the LLM's conditional probability of producing a factually-valid answer. Because our empirical evidence shows that no single rewrite dominates across all linguistic phenotypes, we view the choice as a latent, context-dependent arm selection problem.
> - Contextual bandits are the formal setting for learning a mapping of features to best arms that minimizes expected hallucination probability (proxied by our reward function), without requiring gradients or internal model access. For non-contextual bandits, each algorithm learns a distribution over arms that converge toward the arm / mixture of arms (EXP3) with the highest expected reward.
> - Existence of an optimal policy is guaranteed in bandit literature when rewards are bounded, which holds here because the reward is normalized in $[0, 1]$. Convergence is empirical and consistent with existing theory for Thompson Sampling, LinUCB, and EXP3.
>
> We will clarify that our contribution is a framework-level application of contextual bandits to hallucination mitigation, supported by our reward separability.stability, and regret analyses.
>
> **2. Limited contribution to hallucination mechanisms:** Our paper does not claim to introduce a new mechanistic theory of hallucination formation, rather we contribute a decision-theoretic framework for adapting rewrite strategies based on the linguistic structure of incoming queries. Furthermore, we provide empirical evidence that adaptation exploits predictable, interpretable linguistic correlates of hallucination risk (See fig 5 regression weights). Our method, QueryBandits, enables fully closed-source mitigation without architecture access, which is critical for closed-source models. Our contribution and focus is that adaptive mitigation is crucial when working with LLMs.
>
> **3. Existing hallucination mitigation methods:** We will expand our related-work and discussions to cover more methods, and will discuss some here:
> - Decoding-based methods: These approaches rely on access to token-level probabilities and include methods that exploit logins, entropy sampling, and uncertainty estimation. These are well established for open-source models. However, they are not available for closed-source models.
> - Internal modifications and steering: These require access to the model architecture. Methods include attention head editing, interventions, or logic steering, and fall outside the close-source setting targeted by our method.
> - Retrieval-enhanced: Retrieval methods require an external corpus and while powerful, can show inconsistent gains (Ma 2023, Query Rewriting for Retrieval-Augmented Large Language Models). HotpotQA seems reduced EM (32.36 -> 30.47) in retrieval. Furthermore, RAG based systems often: utilize a considerable portion of the context window, have added latency from retrieval steps, often require reranking models to help disambiguate relevance with cross-encoders. These add considerable compute challenges when at scale.
> - Self-refine incurs a higher token overhead via multiple LLM iterations, but is complementary to QueryBandits. In fact, we designed our formulation so that it could be composed with other methods such as RAG and self-refine, rather than mutually exclusive.
>
> **4. Algorithmic complexity not reported** Here are the details per pipeline step:
> | Stage                 |  Median |    Mean |
> | --------------------- | ------: | ------: |
> | Original Query        |      16 |    19.3 |
> | Feature Tagger Output |     110 |     110 |
> | Rewrite Input + Output      |      44 |    57.4 |
> | Answer Input + Output       |     134 |   249.1 |
> | Judge (Input & Output)           |     162 |   252.3 |
> | Total             | 493 | 688 |
>
> This is approximately $0.0026–$0.0035 per query at current GPT-4o pricing, and represents only 0.5% of 4o's total allowed context. Given our token use per query, we estimate additional cost for 4.1 as 0.0018-0.00265 per query (median-mean), and for gpt-5/5.1: 0.00211-0.00313. Adding judges incurs additional 0.0012 per query. For reference, a baseline cost is 0.0009 per query (no rag, no rewrite).
>
> **5. Title:** We can clarify the title for LLM QA: `No One Size Fits All: QueryBandits for LLM Hallucination Mitigation`
>
> We thank the reviewer and appreciate their recognition of our practicality, experimental validation, and interpretability. We will incorporate the above (+ other comments) into our revision.

---

### Author Response · Authors · 2025-11-27
**First Revision**

Hi everyone, to help ground the revisions we have made in this version:
1. Title and Scope is linked to refer to LLM hallucination mitigation, and we clarified these points in the introduction that our contribution is framework-level rather than a new mechanistic theory of hallucinations.
2. In the abstract we clarified our use of a 17-dim feature vector evaluated on GPT-4o on 16 QA scenarios.
3. We expanded the Action Space section to explain the semantics of each rewrite arm, including that Disambiguate operates purely by rephrasing within the original query (no external knowledge), and all arms instantiated with gpt-4o-2024-11-20 but modular.
4. Remarks on Bandit vs Full RL moved to the appendix.
5. In Reward Design, we clarified our why all 3 terms are used to prevent reward degeneracy. We also added a judge-robustness analysis (Table 6) with callout in the main body.
6. Feature extraction & stability: Added a dedicated Feature Extraction paragraph specifying that we use gpt-4o-2024-11-20 with temperature 0 and structured outputs, and reporting bitwise agreement (~99.3%) and per-feature stability (97.4–99.7%) over 1,000 queries X 5 runs.
7. Introduced a token accounting table in the appendix (as shown in the rebuttals below).
8. Made clarification in the Experiments section on bandit degeneracy under canonical, unperturbed queries leads to arm collapse (consistent with prompt memorization), whereas perturbations induce diverse arm usage and gains, without semantic drift.
9. We clarified that non-contextual bandits often collapse to a single arm per dataset and that contextual TS stochastically dominates EXP3 and static policies in per-query reward, based on new dominance CDFs (Figs 15, 16, 17, 18).

We thank the reviewers so far for their feedback and suggestions. We are still planning to add several more revisions when completed: open-weight baseline, at least one more feature ablation (time permitting), and further polishing of the current revisions in the final version. We hope these clarifications and additions address some of your concerns and make the scope, cost, and robustness of QueryBandits more transparent.

---

### Meta-Review · Area_Chair_9ZMQ · 2026-01-06

**Summary:**

We have three reviewers who have carefully checked this paper and they have diverse recommendations on this paper. The reviewers are mainly concerned about the lack of comparison with methods like RAG, the token or latency cost of the proposed method, and the insufficiency of the ablation studies. While the authors partially addressed these concerns in the rebuttal, a more comprehensive evaluation of the proposed method remains necessary.

**Reviewer Concerns:**

The comparison with existing hallucination mitigation methods is still insufficient, which is the core concern mentioned by all reviewers.

**Reviewer Scores:**

The reviewers may maintain their score due to the insufficient evaluation.

---

### Decision · Program_Chairs · 2026-01-26

Reject